# *Cx3cr1*-deficient microglia exhibit a premature aging transcriptome

Stefka Gyoneva[1] , Raghavendra Hosur[2], David Gosselin[3], Baohong Zhang[2], Zhengyu Ouyang[3], Anne C Cotleur[1] , Michael Peterson[4] , Norm Allaire[2], Ravi Challa[2], Patrick Cullen[2], Chris Roberts[2], Kelly Miao[1], Taylor L Reynolds[4], Christopher K Glass[3,5] , Linda Burkly[1], Richard M Ransohoff[1]

**CX3CR1, one of the highest expressed genes in microglia in mice and humans, is implicated in numerous microglial functions. However, the molecular mechanisms underlying *Cx3cr1* signaling are not well understood. Here, we analyzed transcriptomes of *Cx3cr1*-deficient microglia under varying conditions by RNA-sequencing (RNA-seq). In 2-mo-old mice, *Cx3cr1* deletion resulted in the down-regulation of a subset of immune-related genes, without substantial epigenetic changes in markers of active chromatin. Surprisingly, *Cx3cr1*-deficient microglia from young mice exhibited a transcriptome consistent with that of aged *Cx3cr1*-sufficient animals, suggesting a premature aging transcriptomic signature. Immunohistochemical analysis of microglia in young and aged mice revealed that loss of *Cx3cr1* modulates microglial morphology in a comparable fashion. Our results suggest that CX3CR1 may regulate microglial function in part by modulating the expression levels of a subset of inflammatory genes during chronological aging, making *Cx3cr1*-deficient mice useful for studying aged microglia.**

## Introduction

Microglia are the parenchymal tissue-resident macrophages of the central nervous system (CNS). They support brain development and adult CNS homeostasis, functions which are often dysregulated in acute or chronic neurological conditions (Xavier et al, 2014; Casano & Peri, 2015; Ransohoff, 2016; Hickman et al, 2018; Thion et al, 2018b). Deciphering how microglial phenotypes are regulated and dysregulated is highly relevant for understanding and treating neurodevelopmental and neurodegenerative conditions.

The factors that regulate the microglial transcriptome under homeostatic conditions are starting to emerge. Microglial identity is conferred by their yolk-sac origin and the brain environment (Bennett et al, 2018). The transcription factor Sall1 is essential for the expression of microglia signature genes, which differentiate microglia from other tissue macrophages and myeloid cells (Buttgereit et al, 2016). In multiple conditions, mouse microglia demonstrate a core conserved reactive-state transcriptome, which is extensively modified to yield disease-specific transcriptomes (Chiu et al, 2013; Keren-Shaul et al, 2017; Friedman et al, 2018). Thus, microglia establish and then alter their gene expression and function, based on contextual cues that modulate the activity of an enhancer landscape, primed through the action of lineage-determining transcription factors (Holtman et al, 2017).

The chemokine (C-X3-C) motif receptor 1 (*Cx3cr1*) is highly expressed in microglia. Genetic studies with deletion of one or both copies of *Cx3cr1*, or its monogamous ligand *Cx3cl1*, have shown pleiotropic roles for the receptor. Mice lacking *Cx3cr1* expression show altered brain development during embryogenesis, dysregulated neuronal firing in the first few weeks after birth, and abnormal results in tests designed to mimic sociopsychological behaviors (Paolicelli et al, 2011; Rogers et al, 2011; Zhan et al, 2014; Bolós et al, 2017). Moreover, deletion of *Cx3cr1* often affects outcomes in models of neurodegeneration, with both protective and detrimental effects reported. For example, loss of *Cx3cr1* is detrimental in the acute 1-methyl-4-phenyl-1,2,3,6-tetrahydropyridine (MPTP) model of Parkinson's disease and in tauopathies such as frontotemporal dementia, but promotes amyloid clearance in models of amyloid $\beta$ deposition (Cardona et al, 2006; Bhaskar et al, 2010; Lee et al, 2010; Liu et al, 2010). Both protective and detrimental effects have also been reported for *Cx3cr1* deletion in varied models of traumatic brain injury, including at different time points in the same model (Febinger et al, 2015; Erturk et al, 2016; Zanier et al, 2016). Finally, because of its high and consistent expression, *Cx3cr1* is also often used as a driver in reporter constructs, such as *Cx3cr1-eGFP* mice, or for Cre-mediated gene deletion (Jung et al, 2000; Goldmann et al, 2013; Parkhurst et al, 2013).

[1]Acute Neurology, Biogen, Cambridge, MA, USA    [2]Computational Biology and Genomics, Biogen, Cambridge, MA, USA    [3]Cell and Molecular Medicine, University of California San Diego, San Diego, CA, USA    [4]Translational Neuropathology, Biogen, Cambridge, MA, USA    [5]School of Medicine, University of California San Diego, San Diego, CA, USA

Correspondence: Stefka.gyoneva@biogen.com; Richard_Ransohoff@hms.harvard.edu
David Gosselin's present address is CHU de Québec Research Center, Department of Molecular Medicine, Faculty of Medicine, Université Laval, Québec City, Canada
Richard M Ransohoff's present address is Department of Cell Biology, Harvard Medical School, Cambridge, MA, USA

Despite the importance of CX3CR1 for microglial function in the dynamic contexts of development and disease, there is relatively little understanding of how CX3CR1 regulates microglial function at the molecular level. We performed a transcriptomic analysis of *Cx3cr1*-deficient mice under varying conditions (peripheral immune challenge, across a span of ages and in both genders) to define how transcriptional responses to diverse biological processes were affected by loss of *Cx3cr1*. Our results indicate that CX3CR1 signaling contributes to the regulation of microglial morphology and a subset of inflammatory genes. Importantly, loss of CX3CR1 signaling in young mice resulted in a microglial transcriptome reminiscent of aged mice.

# Results

## *Cx3cr1* deletion alters inflammatory gene expression in microglia from young adult mice

We performed RNA-sequencing (RNA-seq) on isolated brain microglia from young adult (2 mo) mice with variable *Cx3cr1* expression: *Cx3cr1*$^{+/+}$ (WT), *Cx3cr1*$^{+/eGFP}$ (Het), and *Cx3cr1*$^{eGFP/eGFP}$ (KO). In WT mice, the microglial expression profile obtained here is consistent with the published microglial transcriptomes (Table S1) (Hickman et al, 2013; Butovsky et al, 2014; Zhang et al, 2014). To examine the effect of *Cx3cr1* deletion, we initially performed principal component analysis (PCA). The samples separated well by *Cx3cr1* genotype on the first principal component (Fig 1A), showing that *Cx3cr1* expression is a main driver of variability between the samples. In addition, there was a visible separation by gender overall and within each of the genotypes, which was driving the second component of the variability (Fig 1A).

We next identified differentially expressed genes (DEGs) by performing pairwise comparisons between the genotypes. Using the cutoffs described in the Methods, loss of *Cx3cr1* resulted in 165 DEGs. Of these, 100 genes showed lower expression in *Cx3cr1*-KO microglia, whereas 65 genes showed higher expression (Table S2). Similarly, when comparing *Cx3cr1*-Het to WT microglia, there were 55 genes with lower expression in Het animals and only four with higher expression (Table S3). Importantly, most of the genes differentially expressed between Het and WT microglia were also differentially expressed between KO and WT microglia (Fig 1B), indicating that *Cx3cr1*-Het microglia exhibit an intermediate transcriptional profile between WT and KO microglia (see also Fig 1A).

Unsupervised cluster analysis of the top 200 DEGs (by *P*-value) showed a striking resemblance between the transcriptomes of *Cx3cr1*-KO and -Het microglia (Fig 1C). Manual annotation of these DEGs revealed that many of them encoded MHC class II genes (e.g., *H2-Q10*, *H2-Aa*, *H2-Eb1*, and *H2-Ab1*), chemokine receptors and ligands (*Ccr3*, *Cxcr2*, *Ccr5*, *Ccr1*, and *Ccl9*), proteases (*Mmp25*, *Adam8*, *Mmp9*, and *Ctsa*), and other genes with immune function (*Fcrl1*, *S100a8*, *S100a6*, *S100a11*, *Lilra6*, *Lilrb1*, and *Fcrls*). Although fewer genes had higher expression in *Cx3cr1*-deficient microglia, some of these included genes with pleiotropic functions such as *Tnf*, the transcription factors *Egr1* and *Klf2*, and the chemokine-like receptor *Ccr1l1*. We confirmed the altered expression of selected genes by *Cx3cr1* genotype using quantitative reverse transcriptase PCR (qRT-PCR) of

microglia sorted from a separate cohort of mice (Fig S1). Thus, homozygous or heterozygous loss of *Cx3cr1* signaling resulted in dysregulated expression of a subset of genes associated with immune function.

A distinctive transcriptome (termed "signature") differentiates microglia from other myeloid cells (Butovsky et al, 2014). *Cx3cr1* deletion showed only a minor effect on the expression of most signature genes (Fig S2), and only a few microglia signature genes met the criteria for differential expression between *Cx3cr1*-KO and WT microglia.

## Active promoter landscapes do not explain transcriptomic differences in microglia of *Cx3cr1*-deficient mice

We performed chromatin immunoprecipitation for RNA polymerase II (Rbp2) followed by sequencing (ChIP-seq) to determine whether changes in transcriptional activity indicated by Rbp2 binding explain the differential gene expression between *Cx3cr1*-WT and -KO microglia. Unexpectedly, only 19 genes displayed differential binding between WT and KO animals (Fig 2A and Table S4). Moreover, there was no significant correlation between fold changes seen by RNA-seq and ChIP-seq (Fig 2B). These findings were corroborated by minimal changes in open chromatin landscapes assessed by H3K27Ac ChIP-seq (Fig 2C and Table S5).

It is worth noting that many of the genes showing higher Pol–II binding in WT mice appear to be involved in transcription regulation or RNA stability, such as *Klhl42* and *Auh* (Fig 2D). Moreover, *Tnf*, a classical inflammatory gene, had higher expression in *Cx3cr1*-KO microglia by RNA-seq and a trend for differential Rbp2 binding by ChIP-seq (irreproducible discovery rate [IDR] = 0.0553; Fig 2D). In summary, differences between *Cx3cr1*-WT and -KO microglia are likely to be determined by posttranscriptional mechanisms, but transcriptional regulation, particularly for genes whose products regulate mRNA stability and exert wide transcriptomic effects, may also play a small part in the transcriptomic phenotype variation.

## *Cx3cr1* and LPS regulate different transcriptional programs in microglia

Because many of the genes altered by *Cx3cr1* deletion in microglia are immune-related (Fig 1C), we hypothesized that *Cx3cr1* deletion will modulate the transcriptional response of microglia to an acute systemic inflammatory challenge such as i.p. lipopolysaccharide (LPS) injection. The LPS treatment (1 mg/ml, 24 h) induced a strong transcriptional response in both peritoneal cells and in microglia, resulting in 5,362 and 5,084 DEGs, respectively. This was visible by PCA, where LPS and saline samples formed distinct clusters (Fig 3A and B). However, there was no visible effect of *Cx3cr1* genotype on the PCA plots, even when only microglia were examined (Fig 3B). Similar results were obtained with hierarchical clustering analysis (Fig S3). Only 14 genes met our criteria for differential expression by *Cx3cr1* genotype in microglia from LPS-injected mice (Table S6). Yet, some of the genes affected by *Cx3cr1* deficiency in microglia from LPS-injected mice included up-regulation of *Adora1* and *Klf2* and down-regulation of *Ninj1* (Fig 3C), which could mediate subsequent functional effects. Thus, *Cx3cr1* deletion did not strongly affect the transcriptional response of microglia to a single acute systemic LPS treatment.

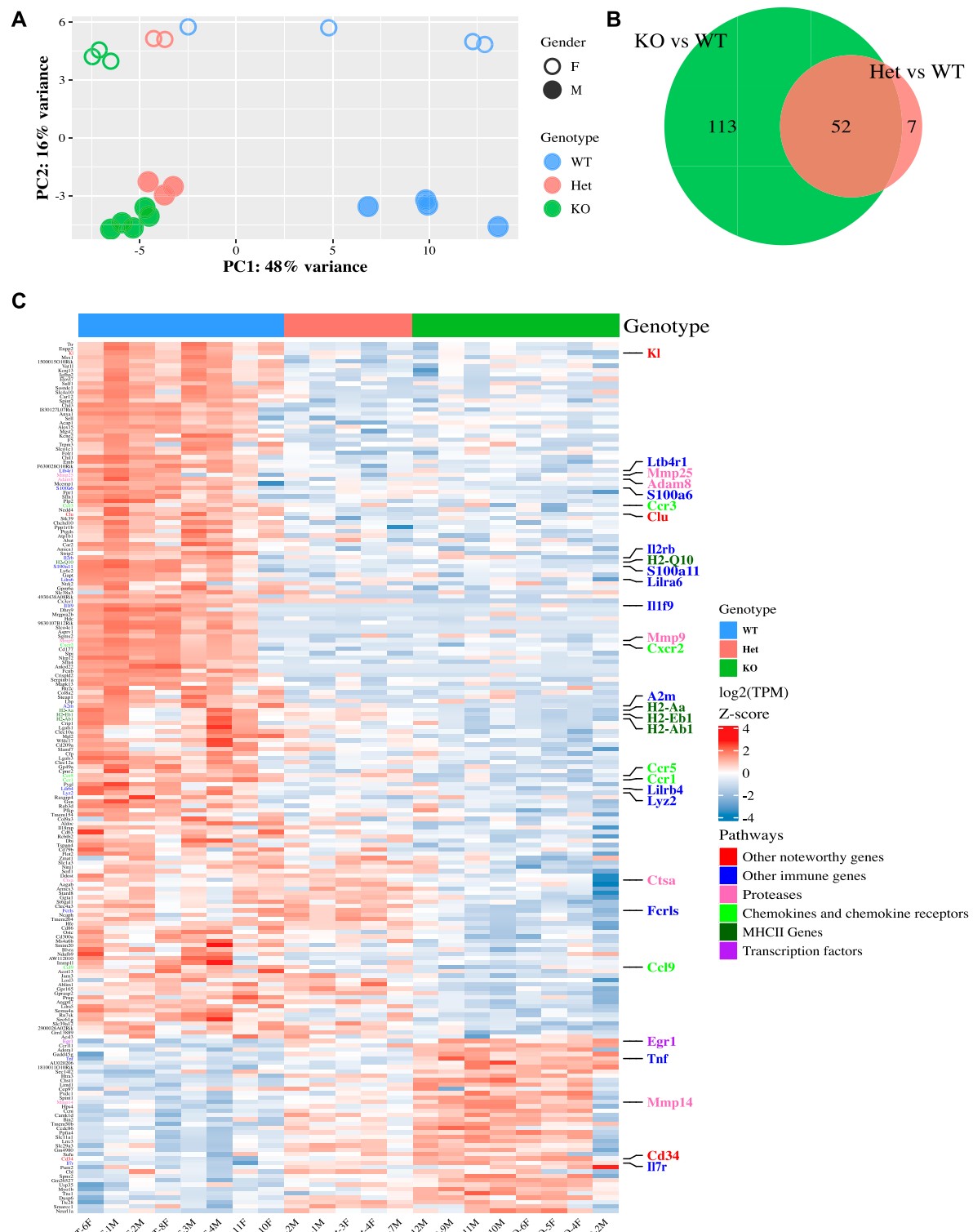

**Figure 1. *Cx3cr1* loss alters the transcriptome of microglia from 2-mo-old mice.**

Microglia were isolated from 2-mo-old *Cx3cr1⁺/⁺* (WT, n = 8), *Cx3cr1⁺/eGFP* (Het, n = 5), or *Cx3cr1eGFP/eGFP* (KO, n = 8) mice. The RNA isolated from microglia of each mouse was used to generate an individual RNA-seq data set. **(A)** The PCA indicates that the samples separate by genotype in the first principal component and gender in the second. Each dot represents microglia from an individual mouse. **(B)** Most of the genes differentially expressed between Het and WT microglia are also differentially expressed by KO and WT microglia. **(C)** Unsupervised cluster analysis of the top 200 DEGs in microglia from 2-mo-old WT (n = 8), Het (n = 5), or KO (n = 8) mice. Most DEGs are expressed at lower levels in Het and KO samples compared to WT samples. Selected genes and the pathways they are associated with (manual annotation) are represented in different colors.

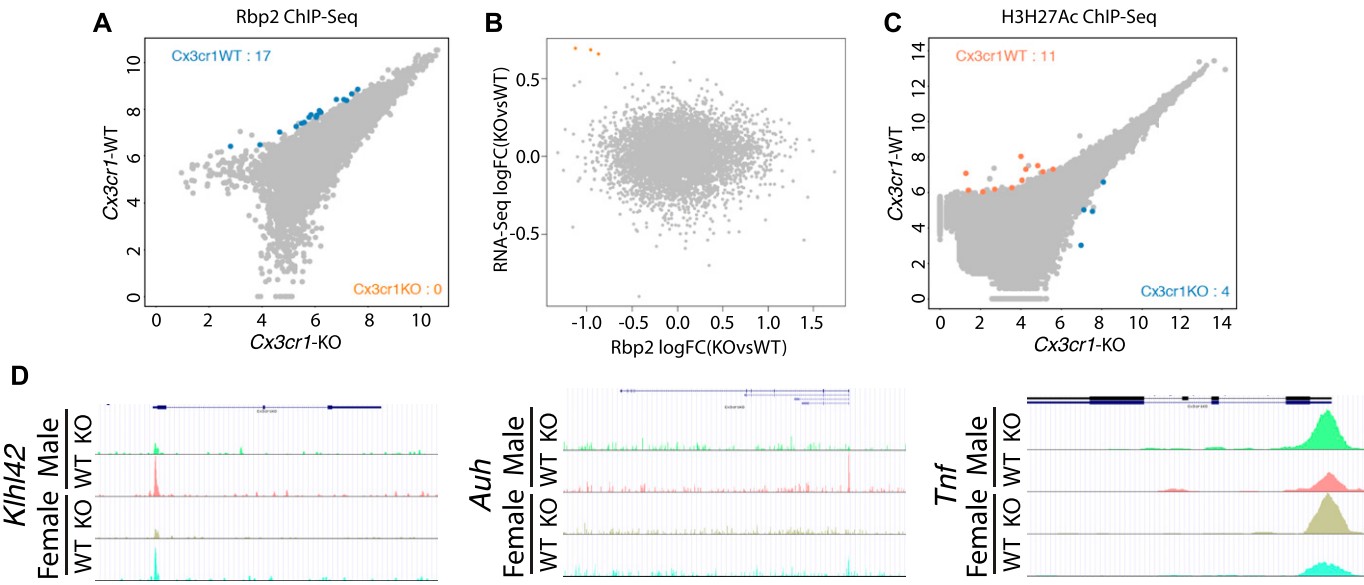

**Figure 2. Active promoter landscape of *Cx3cr1*-KO and –WT samples shows minor differences in Rbp2 binding.**
Microglia were isolated from 2-mo-old *Cx3cr1⁺/⁺* (WT) or *Cx3cr1^eGFP/eGFP^* (KO) mice, pooling brains from three gender-matched mice to obtain samples for chromatin immunoprecipitation followed by sequencing. **(A)** ChIP-seq for Rbp2, a mark of actively transcribed genes, indicates few significant differences between the genotypes. Only genes with at least four sequencing reads in at least one of the genotypes are shown, with genes reaching statistical significance highlighted in blue. **(B)** There is minimal correlation between mRNA expression (fold change determined by RNA-seq) and active transcription (Rbp2 binding determined by ChIP-seq). Genes with fold change > |1.5| are highlighted in coral. **(C)** ChIP-seq for H3K27Ac, a mark of open and active chromatin, indicates few significant differences between the genotypes. Only genes with at least four sequencing reads in at least one of the genotypes are shown, with genes reaching statistical significance highlighted in blue and coral. **(D)** The UCSC browser snapshots of Rbp2 binding for selected genes. Exons are represented by boxes, and direction of transcription is indicated by arrows in introns.

## The transcriptome of young *Cx3cr1*-deficient mice resembles that of aged WT mice

Age is the dominant risk factor for neurodegenerative diseases and also alters the microglial transcriptome both in mice and humans (Grabert et al, 2016; Olah et al, 2018). To examine how *Cx3cr1* deficiency affects microglial transcriptomes during aging, we isolated microglia from young adults (2 mo), middle-aged (1 yr), and aged (2 yr) *Cx3cr1⁺/⁺*, *Cx3cr1⁺/eGFP*, and *Cx3cr1^eGFP/eGFP^* mice. Fewer live microglia were recovered from aged mice, independent of genotype (Fig 4A). There was a visible separation by age in the PCA (Fig 4B), highlighting the importance of age in regulating microglial transcriptome. In microglia from WT mice, we detected 1,082 DEGs between 1-yr- and 2-mo-old mice, 2,046 DEGs between 2-yr- and 2-mo-old mice, and the signature was consistent with previous studies (Tables S7 and S8). By contrast, KO microglia showed 706 DEGs between 1-yr- and 2-mo-old mice and 1,312 DEGs between 2-yr- and 2-mo-old mice (Tables S9 and S10). The genotype difference appeared to grow smaller with age, being less detectable by PCA in aged mice (Fig 4B). When 2-yr-old mice were examined, the separation by genotype was still evident (Fig 4C) despite a reduced number of genotype-regulated DEGs (Tables 1 and S11). It is worth noting that there was substantial overlap in the age-dependent DEGs between WT and KO microglia (Fig 4D).

We hypothesized that *Cx3cr1* deletion might modify the transcriptome of microglia from younger mice to resemble that of microglia from aged mice. To address this possibility, we performed unsupervised cluster analysis (Fig 5). In microglia from 2-mo-old mice, the samples separated by genotype, with Het mice displaying an intermediate gene expression between WT and KO mice, recapitulating our earlier results (Fig 1C). In older mice, WT and Het samples from both 1-yr- and 2-yr-old mice were grouped together, and KO samples from 1-yr- and 2-yr-old mice were grouped together. At these older ages, WT and Het appeared more similar than either of the genotypes to KO.

Importantly, there was a large cluster of genes—all of which were down-regulated compared with young WT microglia—that showed very similar expression between 2-mo-old Het and KO microglia and older mice of any genotype. Genes within this category included many genes that we had already identified as down-regulated by *Cx3cr1* deletion in 2-mo-old mice, such as MHC class II (*H2-Q10, H2-Aa, H2-Eb1,* and *H2-Ab1*), chemokine receptors (*Ccr3* and *Ccr5*), *Clu, Mmp9,* and *Lyz2*. Thus, the absence of a single copy of *Cx3cr1* at young age results in a transcriptome comparable with that of aged microglia. Additional clusters of genes were up-regulated with aging but showed highest expression in aged KO microglia. Noteworthy genes in these clusters included *Adora1, Cd34, H2-Dma,* and *Tnf*. The widespread dysregulation of immune genes by *Cx3cr1* deletion was also confirmed by gene ontology analysis (Table 2).

We also performed unsupervised cluster analysis using the top 200 DEGs between WT 2-yr versus 2-mo microglia rather than the genotype-regulated DEGs (Fig S4). Using this gene list, the samples clearly clustered by age. Genes in the pathways regulated by *Cx3cr1* loss, namely MHCII genes, chemokines and chemokine receptors, and inflammatory genes were also regulated by age, as has been shown before (Grabert et al, 2016). Moreover, although the clustering by genotype was not as obvious within this aging transcriptome, there

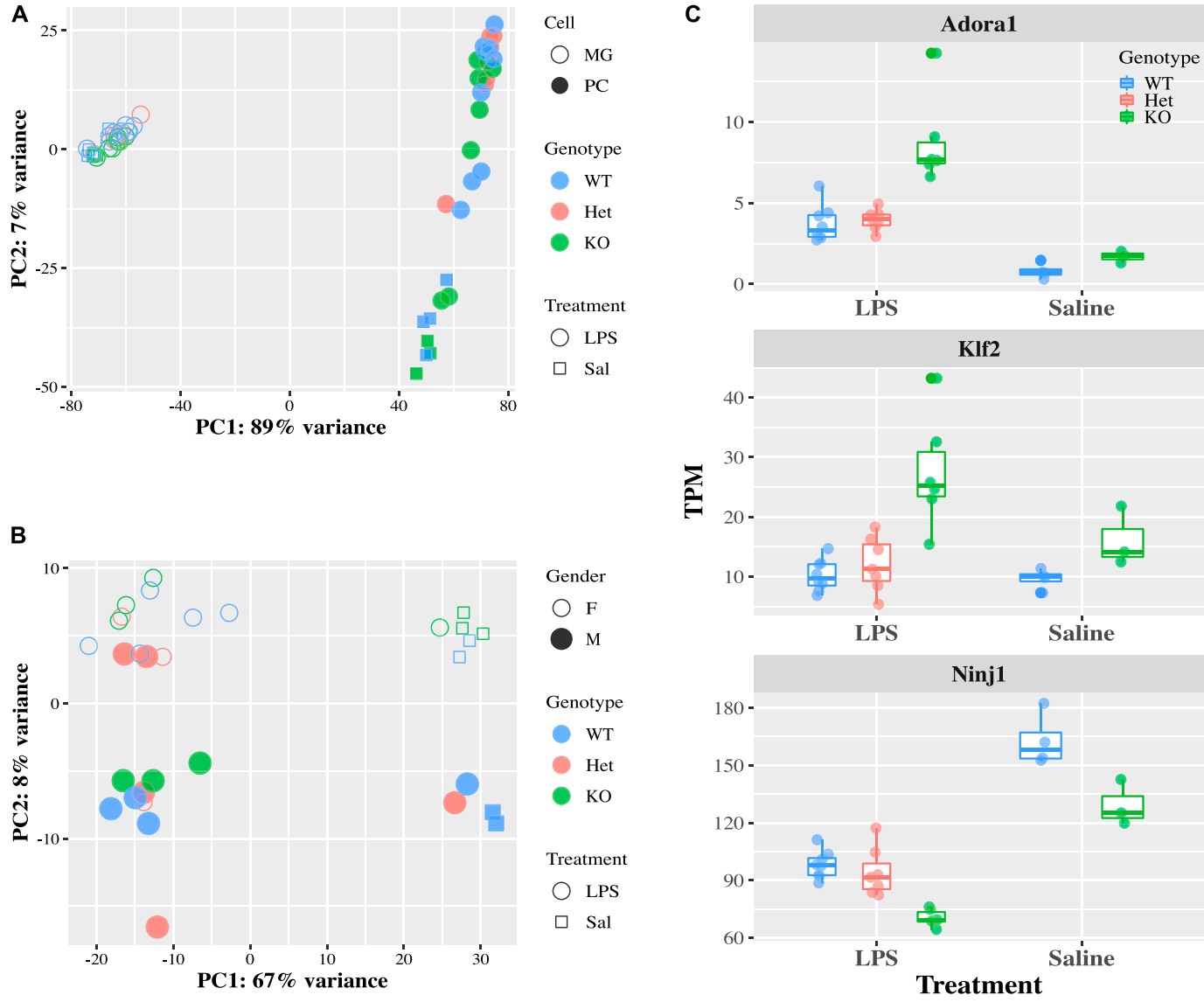

**Figure 3. Loss of *Cx3cr1* does not affect microglial transcriptional response to LPS.**
2-mo-old *Cx3cr1*[+/+] (WT, n = 7), *Cx3cr1*[+/eGFP] (Het, n = 6), or *Cx3cr1*[eGFP/eGFP] (KO, n = 6) mice were injected with 1 mg/kg LPS i.p. (or saline control, four WT and three KO animals only). Twenty-four hours later, peritoneal cells were collected, and microglia were sorted from isolated brains. Both cell populations were used for RNA-seq. **(A)** PCA of all samples. Each dot within a cell type represents an individual animal, but there are matched microglia and peritoneal cells from the same animal. Samples, coded based on the cell type (open or solid fill), genotype (color), and treatment (symbol), separate by cell type and treatment. **(B)** The PCA of microglia only shows separation by treatment, but not genotype in LPS-injected mice. Samples are coded by gender (open or solid fill), genotype (color), and treatment (symbol). **(C)** Expression of selected genes that are modulated by *Cx3cr1* genotype after LPS injection.

appeared to be a grouping of WT and Het samples in 2-mo-old mice, which was lost on the aged mice. These results support our finding that *Cx3cr1* loss and aging modulate the expression of similar genes and pathways.

Although sample separation by gender was evident at all time points (Fig 4B), it was especially visible in microglia from 2-yr-old mice where it drove some of the primary variability component (Fig 4C). Indeed, the number of gender-regulated DEGs increased with age, peaking at 1 yr, but remained high in microglia from 2-yr-old mice (Tables 1 and S12). In microglia from 2-yr-old mice, almost all of the gender-biased DEGs showed higher expression in females than in males, regardless of *Cx3cr1* genotype, so that the samples primarily

clustered by gender, but not by genotype (Fig 6). These gender-biased DEGs included many genes with immune functions, such as *Csf1*, *Nfkb2*, *Tlr2*, *Cxcl16*, *Cxcl14*, *Itgax*, *Ccl4*, *Ccl3*, and *Tlr7*, all of which are different from the genotype- and age-regulated DEGs mentioned above. The validity of the analysis was confirmed by showing differential expression of the sex chromosome genes (Fig 6).

### *Cx3cr1* deletion alters microglial morphology in young mice

Microglial morphology is often used to infer information about microglial reactivity. Thus, we extended our analysis by examining whether *Cx3cr1* deficiency affected microglial morphology in young

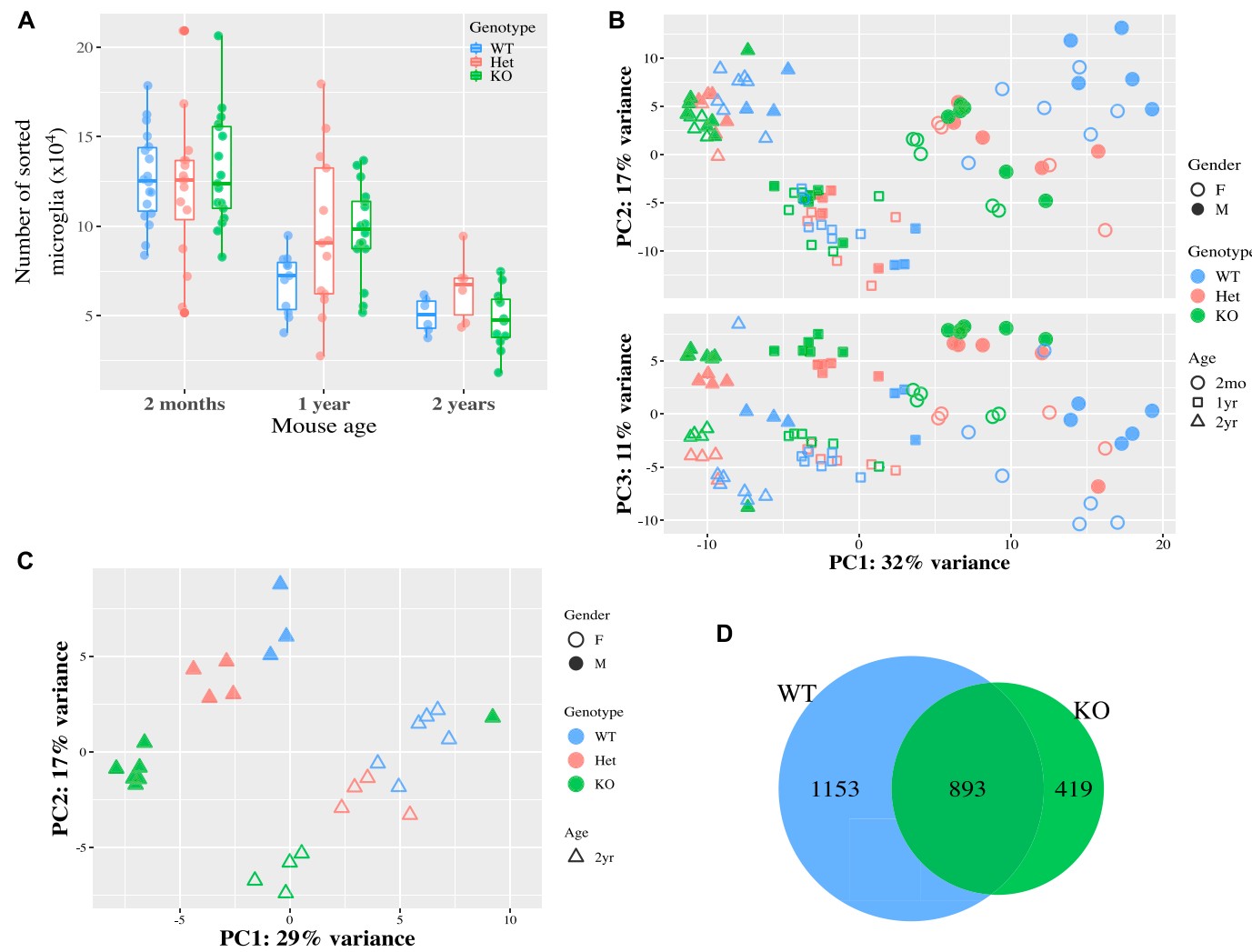

**Figure 4. Effect of aging and *Cx3cr1* deletion on microglial transcriptome.**
Microglia were isolated from 2-mo-, 1-yr-, or 2-yr-old *Cx3cr1*<sup>+/+</sup> (WT), *Cx3cr1*<sup>+/eGFP</sup> (Het), or *Cx3cr1*<sup>eGFP/eGFP</sup> (KO) mice. **(A)** Fewer microglia were recovered from 2 yr mice, but the recovery was independent of genotype. **(B, C)** RNA-seq was performed on all samples together. Each symbol represents microglia from an independent animal. **(B)** The PCA for all samples suggests separation by age as the primary factor driving variability, with genotype second and gender third (B). **(C)** For microglia from 2-yr-old mice (C), the samples separate by genotype and by gender within each genotype. **(D)** Overlap of DEGs between 2-yr and 2-mo microglia in WT and in KO mice. There is a large amount of shared DEGs between the genotypes.

(2 mo) or old (18–24 mo) mice by quantitative morphometry, using immunohistochemistry for Iba1, a cytosolic marker, to delineate the cells. Microglia in the cortex (Ctx) (Fig 7A), hippocampus (HC) (Fig S5A), and striatum (Str) (Fig S5B) displayed a ramified morphology across all ages and genotypes, in both male and female mice. By contrast, cerebellar (Cbm) microglia appeared less ramified than in other brain regions in both genotypes (Fig 7B).

We next quantified the microglial cell number and morphology in the various brain regions. Because there were only modest differences (Fig S6), we pooled the genders for analysis. The microglial number was not significantly affected by genotype in the cortex, striatum, or HC (Figs 7C and S5C and D). With respect to morphology in the cortex, young *Cx3cr1*-deficient microglia had significantly decreased total Iba1 area, Iba1 area in processes directly connected to cell bodies, and process perimeter compared with those found in WT mice (Fig 7C), which could be explained by thinner microglial

processes in young KO mice. There was a significant effect of age for the microglial process area between young and old mice, but the genotypes were no longer different in aged mice (Fig 7C), which is consistent with the reduced number of transcriptional changes between aged WT and KO microglia. Similar results were also detected in the HC and striatum: young KO microglia generally showed reduced Iba1 process area and/or perimeter, which increased with age, but these patterns were lost in the aged microglia (Fig S5C and D).

In the cerebellum (Fig 7D), there was a significant increase in microglial numbers in aged mice in both genotypes. Microglial morphology accessed by the total Iba1 area and cell body–associated Iba1 dendritic process area and perimeter in the cerebellum was much lower than in the cortex, consistent with less ramified microglia in this brain region (Ayata et al, 2018). The cerebellar process area and perimeter were significantly increased in old mice of either

**Table 1.  Number of differentially regulated genes by age.**

| Comparison | 2 mo | | 1 yr | | 2 yr | |
|---|---|---|---|---|---|---|
| | Up | Down | Up | Down | Up | Down |
| KO versus WT | 65 | 100 | 23 | 6 | 3 | 14 |
| F versus M[a] | 4 | 5 | 66 | 6 | 42 | 2 |

[a]WT mice only.
Only genes with FDR < 0.05, fold change > 30%, and average expression > 4 TPM were considered.

genotype (Fig 7D), but were not different between the genotypes. This finding was mainly attributable to dense immunoreactivity around the nucleus of reactive-appearing, amoeboid cells in both genotypes (Fig 7B). Interestingly, clusters of microglia were readily detectable in the cerebellum of old mice, except for male Cx3cr1-WT mice (Fig 7B).

# Discussion

Here, we analyzed how Cx3cr1 deletion affects microglial transcriptome and morphology. We characterized the effects of Cx3cr1 deficiency in multiple contexts, including gender, inflammatory challenge with i.p. LPS, and aging. We show that in microglia from young mice, loss of Cx3cr1 alters the expression of a subset of immune-related genes, with most being down-regulated. Although inflammatory challenge with LPS elicits transcriptional changes in microglia, there is very little overlap with the changes caused by Cx3cr1 deletion. Notably, aging and Cx3cr1 modulate the expression of a similar set of genes, most of which are down-regulated, with Cx3cr1 deficiency accelerating the temporal trajectory of change. Finally, histological analysis of microglia showed age-dependent alteration of microglial morphology in Cx3cr1-deficient mice.

## Accelerated aging transcriptome mediated by deficient Cx3cr1 signaling in Cx3cr1^{eGFP/+} and Cx3cr1^{eGFP/eGFP} mice

Aging is the most salient and universal risk factor for virtually all neurodegenerative diseases. Microglia undergo substantial changes in gene expression and morphology with age (Poliani et al, 2015; Grabert et al, 2016; Olah et al, 2018), but it is not known whether the changes in microglia occur as a consequence of aging or help drive the process. Interestingly, the genotype-driven transcriptional changes we saw between Cx3cr1-KO versus WT and Cx3cr1-Het versus WT microglia at 2 mo of age closely resembled the age-related variation seen between microglia from 2-mo- and 2-yr-old WT mice (Figs 4 and 5). Importantly, the gene expression changes we describe here are consistent with those reported in aged human microglia (Olah et al, 2018).

It has been reported that CX3CL1 mRNA and protein expression decrease in the aged rodent brain (Lyons et al, 2009; Bachstetter et al, 2011), suggesting that reduced CX3CL1-CX3CR1 signaling may contribute to the transition of microglia to an aged phenotype. Indeed, Kl (Klotho) is down-regulated in Cx3cr1-deficient microglia, and Klotho protein has been linked to aging and homeostatic support of brain function (Kuro-o et al, 1997). In addition, lack of Cx3cr1 expression reduces adult neurogenesis in the HC and

olfactory bulb, and impairs the integration of newborn neurons (Bachstetter et al, 2011; Rogers et al, 2011; Xiao et al, 2015; Sellner et al, 2016; Reshef et al, 2017). Similar changes in neurogenesis occur in aging as well, and are partially reversed by exogenous administration of CX3CL1 (Bachstetter et al, 2011), suggesting that the altered microglial function in Cx3cr1-KO mice impairs neurogenesis.

The finding that Cx3cr1 deletion induces similar transcriptional changes in microglia as does aging provides cues to the potential function of CX3CR1. Its activity could be required to maintain homeostatic microglial functions associated with healthy young adult microglia. In this context, deficient Cx3cr1 signaling may provide a useful surrogate for microglial phenotypes associated with aging. The use of Cx3cr1-KO animals as an "isochronic" aging model may be advantagous over established senescence models in that it avoids the widespread systemic changes seen in other models and can help isolate potential roles of microglia.

## Haploinsufficiency of Cx3cr1 loss

Using Cx3cr1-eGFP mice (Jung et al, 2000), we determined that loss of a single copy of Cx3cr1 alters the microglial transcriptome at 2 mo of age. Virtually all genes that show differential expression between microglia from Het and WT mice are also significantly different between KO and WT mice (Fig 1B), and some DEGs showed intermediate expression in Het microglia compared with WT and KO (Fig 1C). Thus, microglia heterozygous for Cx3cr1 exhibit functional haploinsufficiency.

Manual annotation of DEGs suggests that antigen presentation in response to pathogen invasion and chemokine–chemokine receptor signaling were strongly affected by loss of Cx3cr1 (Fig 1C), which was confirmed by gene ontology analysis (Table 2). These findings were surprising because deletion of Cx3cr1 consistently enhanced inflammatory responses in CNS disease models (Cardona et al, 2006; Bhaskar et al, 2010; Lee et al, 2014). Reassuringly, our results with isolated microglia were consistent with analysis of gene expression in hippocampal lysates from WT or Cx3cr1-deficient mice (Rimmerman et al, 2017). Taken together, results of our present transcriptomic analysis, in the context of previous functional studies of Cx3cr1 deficiency in development, disease models, and aging, suggest that examination of the "usual suspects" among myeloid innate-immune genes that exert prominent functions in the periphery carries little explanatory power for attributes of microglia in the intact CNS.

Many investigators use Cx3cr1^{+/−} mice as "wild type" controls for their studies, in part because of the convenience of the reporter and also because Cx3cr1 regulatory elements are used to drive Cre recombinase for gene targeting (Jung et al, 2000; Goldmann et al, 2013; Parkhurst et al, 2013). In CNS disease models, mice deficient for only one copy of Cx3cr1 will unpredictably either show phenotypes similar to WT mice (Fumagalli et al, 2013) or intermediate between WT and KO (Lee et al, 2010; Rogers et al, 2011). It is worth noting that the effect of loss of Cx3cr1 changes with activation or age as WT and Het samples cluster closer to each other in old mice (Fig 5). Our data indicate that Cx3cr1 genotype should be taken into consideration during experimental design.

An important caveat of our study is that we used transgenic mice that are germline and globally deficient for Cx3cr1. Although cellular expression of Cx3cr1 is highest for microglia, there is also robust Cx3cr1 expression by varied subsets of peripheral immune cells and other CNS-associated macrophages (Jung et al, 2000).

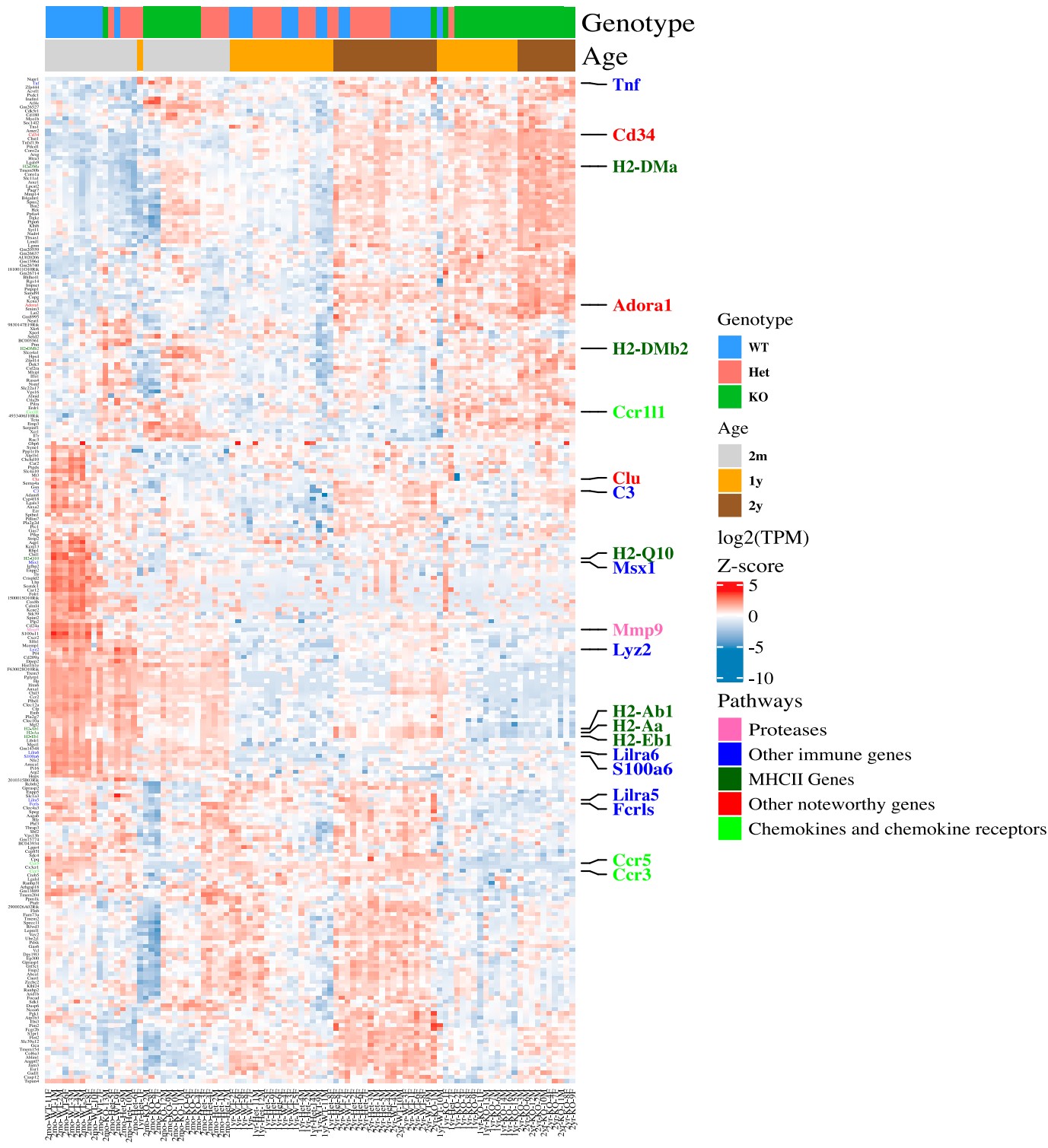

**Figure 5. Young *Cx3cr1*-deficient mice display a premature aging phenotype.**
Microglia were isolated from 2-mo-, 1-yr-, or 2-yr-old *Cx3cr1*⁺/⁺ (WT), *Cx3cr1*⁺/ᵉᴳᶠᴾ (Het), or *Cx3cr1*ᵉᴳᶠᴾ/ᵉᴳᶠᴾ (KO) mice and used for RNA-seq analysis, followed by quantification and unsupervised hierarchical clustering for the top 100 DEGs (KO versus WT, by *P*-value) at each time point, resulting in a combined list of 254 genes. In 2-mo-old mice, the samples separate by genotype, whereas in older mice, they first separate by age and then genotype. Overall, 2-mo-old *Cx3cr1*-Het and KO microglia resemble aged mice of any genotype. Some genes increase with both age and *Cx3cr1* deletion and show the highest expression in aged *Cx3cr1*-KO microglia.

**Table 2.  Gene ontology pathways common to DEGs overtime.**

| | Pathway | 2 mo | | 1 yr | | 2 yr | |
|---|---|---|---|---|---|---|---|
| | | No. of up | No. of down | No. of up | No. of down | No. of up | No. of down |
| 1 | Immune response | 14 | 34 | 8 | 9 | 12 | 11 |
| 2 | Transcription factor binding | 27 | 5 | 4 | 2 | 4 | 19 |
| 3 | Regulation of immune response | 11 | 27 | 9 | 4 | 12 | 9 |
| 4 | Regulation of response to external stimulus | 11 | 25 | 5 | 8 | 6 | 14 |
| 5 | Positive regulation of the immune system process | 16 | 28 | 10 | 4 | 12 | 11 |
| 6 | External side of the plasma membrane | 3 | 20 | 5 | 6 | 5 | 12 |
| 7 | Leukocyte migration | 5 | 16 | 1 | 5 | 3 | 6 |
| 8 | Vacuole | 15 | 14 | 9 | 6 | 15 | 8 |
| 9 | Regulation of cell activation | 9 | 24 | 7 | 6 | 10 | 9 |
| 10 | Lysosome | 14 | 14 | 8 | 6 | 13 | 6 |
| 11 | Cell migration | 18 | 26 | 7 | 5 | 9 | 16 |
| 12 | Response to wounding | 14 | 27 | 7 | 6 | 12 | 11 |
| 13 | Endosome | 15 | 24 | 5 | 9 | 12 | 15 |
| 14 | Carbohydrate binding | 4 | 23 | 2 | 1 | 3 | 7 |
| 15 | Leukocyte chemotaxis | 2 | 12 | 0 | 4 | 2 | 5 |
| 16 | Regulation of lymphocyte activation | 8 | 20 | 7 | 5 | 8 | 7 |
| 17 | Regulation of defense response | 9 | 22 | 2 | 5 | 4 | 10 |
| 18 | Regulation of locomotion | 14 | 21 | 7 | 6 | 7 | 14 |
| 19 | G-protein–coupled peptide receptor activity | 2 | 11 | 2 | 4 | 2 | 5 |
| 20 | Regulation of cell migration | 13 | 19 | 7 | 6 | 7 | 14 |

Top 20 pathways affected by *Cx3cr1* genotype (KO versus WT) are shown, with the number of genes up-regulated or down-regulated at each time point. Pathways showed an overall increase (tan highlight) or decrease (blue highlight) in expression. Only genes with FDR < 0.05, fold change > 30%, and average expression > 4 TPM were included in the enrichment analysis.

Furthermore, peripheral CX3CR1 can affect brain processes, such as cognition and memory (Garré et al, 2017). CX3CR1 is involved in intestinal macrophage sensing of the microbiota (Niess et al, 2005), and gross disruption of the microbiota can affect microglial transcriptomes (Erny et al, 2015; Thion et al, 2018a). In light of these considerations, the microglial transcriptomic phenotypes observed in *Cx3cr1*-deficient mice could be governed in part by loss of peripheral *Cx3cr1* or loss of *Cx3cr1* in the CNS-associated macrophages. Mice that enable conditional deletion of *Cx3cr1* are not currently available.

Moreover, we cannot rule out the possibility that loss of *Cx3cr1* may affect our ability to recover microglia from all conditions with equal efficiency. As such, the transcriptomes we report here may not be generalizable to the whole microglial population in situ in the brain. Yet, both histology (Fig 7) and RNA-seq analysis (Fig 1) suggest that microglia from *Cx3cr1*-deficient animals show a reduced inflammatory response, supporting the validity of our observations.

## Female gender and peripheral LPS modulate immune gene expression in microglia independently of *Cx3cr1* genotype

Although gender exerted an effect on gene expression in microglia from young adults (2-mo-old mice, Fig 1A), it was even more notable in microglia from 2-yr-old mice by PCA (Fig 4C), long after reproductive senescence. Interestingly, gender-biased genes in old

mice were related to immune function, but were distinct from those immune genes regulated by *Cx3cr1* genotype. Microglial immune genes typically showed higher expression in females (Fig 6 and Table S10). In this regard, a sexually dimorphic microglial development index has been calculated, based on genes that increased in expression during development from E18 to P60 (Hanamsagar et al, 2017). Our results confirm and extend this concept: first, microglial immune gene expression changes during development (Hanamsagar et al, 2017) and in aging (this study); second, females show higher expression of immune-related pathways. Considering common gender bias in the incidence of neurodevelopmental and neurodegenerative conditions and the emerging importance of gender differences in the role of microglia in response to multiple perturbations (Sorge et al, 2015; Hanamsagar et al, 2017; Thion et al, 2018a), an in-depth examination of gender differences is warranted, but beyond the scope of this study.

In this study, *Cx3cr1* genotype did not affect the microglial response to peripheral LPS (Fig 3 and Table S5). It should be noted that daily injections of i.p. LPS for 4 d induce more neuronal death in *Cx3cr1*$^{eGFP/eGFP}$ mice than in *Cx3cr1*$^{+/eGFP}$ mice (Cardona et al, 2006). This LPS injection paradigm differs from the one we used and generates a different activation status in microglia (Wendeln et al, 2018). Although *Cx3cr1* is dispensable for the microglial transcriptional response to a moderate dose of LPS at 24 h after injection, the

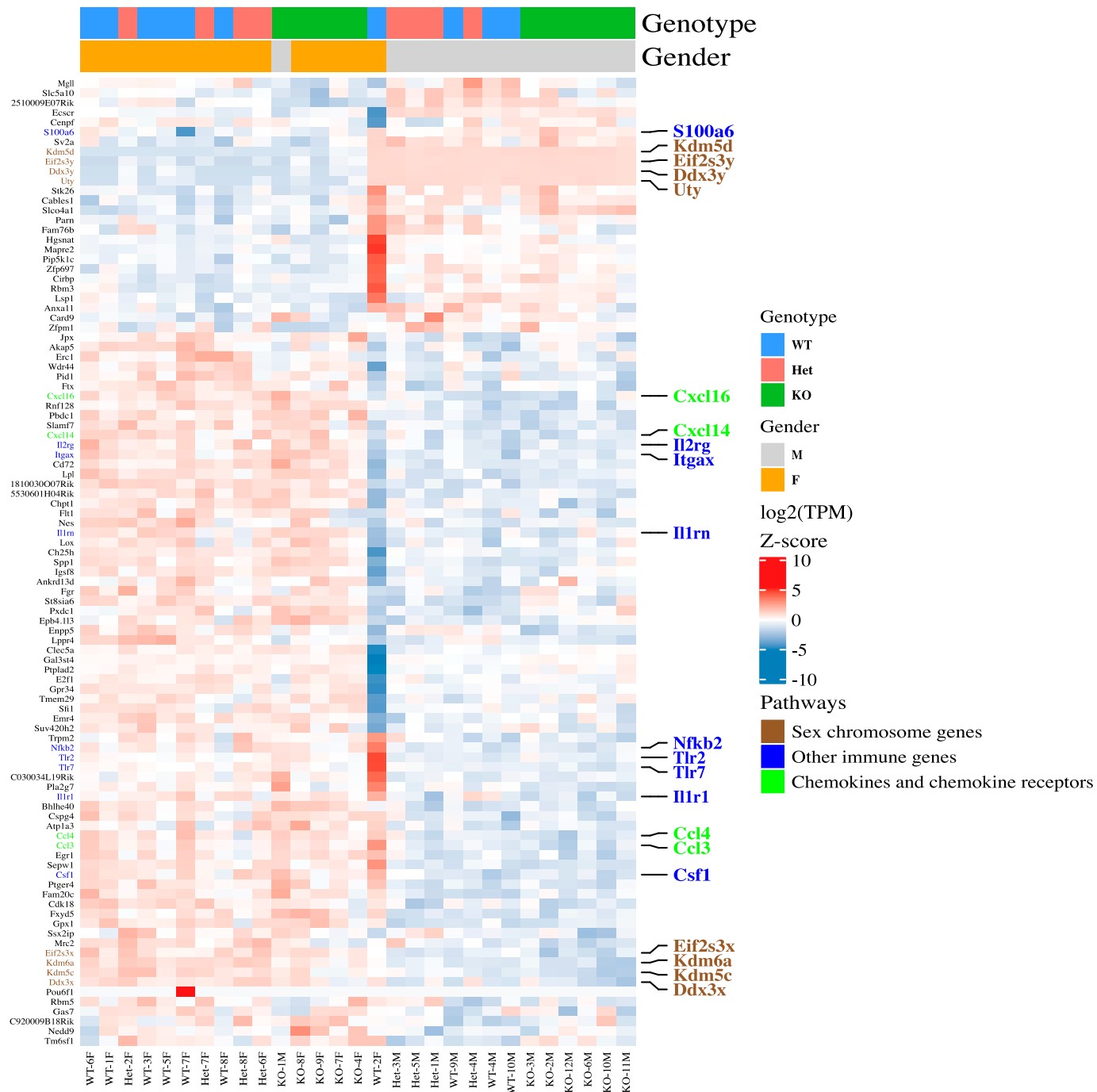

**Figure 6. Increased inflammatory gene expression in microglia from 2-yr female mice.**
Quantification and unsupervised hierarchical clustering for microglia from 2-yr-old *Cx3cr1*[+/+] (WT, n = 9), *Cx3cr1*[+/eGFP] (Het, n = 8), or *Cx3cr1*[eGFP/eGFP] (KO, n = 11) mice highlights a strong effect of gender on gene expression in old mice. The samples primarily separate by gender and not *Cx3cr1* genotype. Selected genes and associated pathways (manual annotation) are represented in different colors.

augmented up-regulation of *Klf2*, among other genes, could mediate durable functional responses by microglia (Das et al, 2006; de Bruin et al, 2016; Roberts et al, 2017). In addition, *Cx3cr1*-KO microglia failed to up-regulate the adhesion molecule *Ninj1* after LPS, which could play a role in the neuronal cell death after recurrent systemic LPS treatment (Araki & Milbrandt, 1996; Cardona et al, 2006).

## Loss of *Cx3cr1* results in altered microglial morphology in young mice

We extended our characterization of the effects of *Cx3cr1* deletion on microglia by analyzing the morphology of Iba1-immunoreactive cells (Fig 7). Microglia in the cortex, striatum, and HC of young mice showed

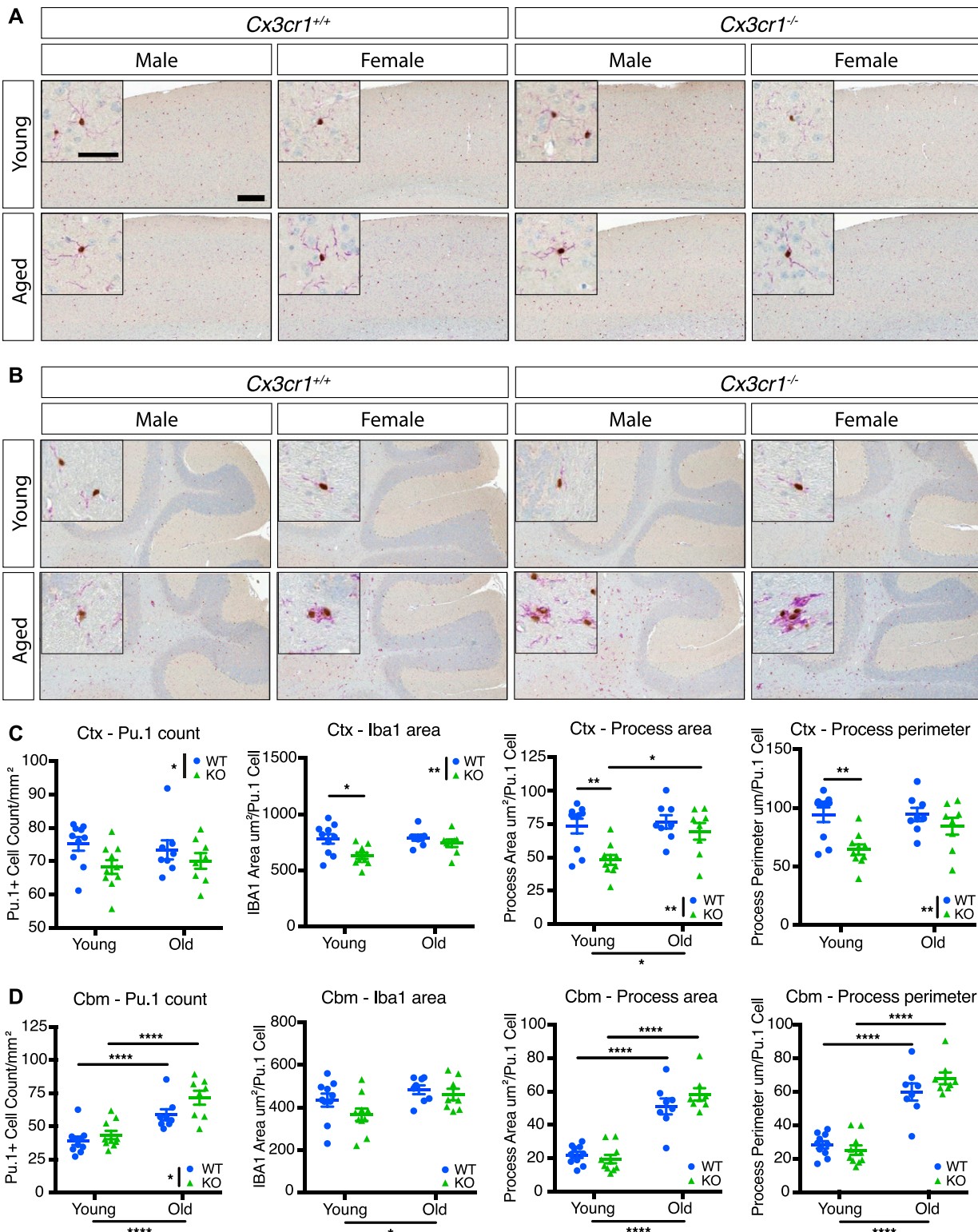

**Figure 7. *Cx3cr1* deletion alters microglial morphology in young and aged mice.**
**(A, B)** Representative images of brain sections from the cortex (A) or cerebellum (B) of young (2 mo) or old (18–24 mo) *Cx3cr1*$^{+/+}$ or *Cx3cr1*$^{eGFP/eGFP}$ male or female mice immunoreacted with anti-Iba1 (purple) and anti-Pu.1 (brown). Note the overall different microglial morphology in the cerebellum in both young and old mice compared with the cortex. Many microglial clusters were detectable in the cerebellum of old animals of either genotype. Scale bar: large image, 200 $\mu$m; inset, 50 $\mu$m. **(C, D)** Quantification of microglial cell numbers and morphology in the cortex (C) and the cerebellum (D). Microglial cell numbers were quantified as the number of Pu.1-positive nuclei per unit area. Microglial morphology was assessed as the total Iba1 immunoreactive area per Pu.1$^{+}$ nucleus, dendritic process area per Pu.1$^{+}$ nucleus, and

reduced ramification assessed by the reduced Iba1 process area and perimeter in *Cx3cr1*-KO mice compared with -WT mice (Figs 7A and C and S5). Lower microglial ramification is often considered a sign of homeostatic microglia or reduced reactivity, which is consistent with the reduced expression of inflammatory genes we observed by RNA-seq (Fig 1). In addition, we detected an age-associated increase in the Iba1 process area and/or perimeter in most brain regions (Figs 7C and D and S5C and D), consistent with a more inflammatory microglia at this time point (Grabert et al, 2016).

Although morphology does not exhibit a direct relationship to defined microglial transcriptomes, our observations suggest that transcriptomes can correlate with other physiological attributes. Specifically, both RNA-seq and immunohistochemistry detected a difference between *Cx3cr1*-WT and -KO microglia in young mice.

## Conclusions

Only by examining the effect of *Cx3cr1* loss under multiple conditions, we were able to understand the interaction of multiple factors in shaping microglial transcriptomes. Here, we show that loss of *Cx3cr1* alters the microglial transcriptome, especially affecting the expression of certain immune pathways (MHC II genes, chemokines and chemokine receptors, and transcriptional factors). The changes we detected in young *Cx3cr1*-KO versus *Cx3cr1*-WT mice were strikingly similar to those seen in *Cx3cr1*-WT aged versus young mice, suggesting that loss of *Cx3cr1* accelerated microglial aging.

# Materials and Methods

## Mice

All procedures were reviewed and approved by the Institutional Animal Care and Use Committee at Biogen. *Cx3cr1*-deficient mice, in which both copies of *Cx3cr1* are disrupted by the gene for *eGFP* (Jung et al, 2000), were initially obtained from The Jackson Laboratory and then bred to maintain them on the C57Bl/6J background. All mice used in the current manuscript are littermate controls derived from *Cx3cr1*$^{+/eGFP}$ crosses. The mice were maintained on a normal diet in a 12-h light/dark cycle. Approximately five mice were used per group (per gender, age, genotype, and treatment). Generally, microglia isolated from individual mice were treated as individual samples. For ChIP-seq analysis, microglia from three mice (gender-matched) were pooled to obtain individual samples.

For some experiments, mice were injected with 1 mg/kg LPS derived from *Escherichia coli* strain O111:B4 and euthanized 24 h later for peritoneal cell and microglia isolation. Control mice received saline injections.

## Microglia isolation

Except for mice used in the LPS study, mice were deeply anesthetized with 10 mg/kg ketamine, 30 mg/kg xylazine until they failed to respond to toe pinch, and transcardially perfused with cold PBS. Mice for the LPS study were euthanized by $CO_2$ asphyxiation, peritoneal cells were isolated by flushing the peritoneal cavity with cold PBS, and then perfused. In all cases, the brain was then isolated and stored in PBS on ice until processing. Brains were processed in sets of 5–7 per day; each day contained a mixture of genotypes and experimental conditions (e.g., saline/LPS). Sequencing results showed no batch effects based on processing days (data not shown).

All subsequent steps were performed on ice. After isolation, whole brains were finely minced with a razor blade and transferred to 15-ml polypropylene conical tubes containing 3 ml accutase enzyme (SCR005; Millipore). The samples were incubated for 30 min at 4°C to allow tissue dissociation. This was followed by manual pipetting consecutively with 5-ml serological pipette and a 1-ml pipette tip to complete the homogenization. The supernatant containing single-cell suspension was passed through a 250-$\mu$m filter in a clean 15-ml conical tube, and the tubes were filled with HBSS supplemented with 25 mM Hepes and centrifuged at 600$g$ for 5 min. The cell pellet was resuspended in FBS, and the volume was adjusted to 10 ml with isotonic 33% Percoll (17-0891-01; GE Healthcare) and centrifuged at 800$g$ at room temperature for 15 min with no brake. The myelin layer was first aspirated from the top, and then the isolated cells collected from the bottom of the tube. The cells were washed in FACS buffer (HBSS containing 1% bovine serum albumin, 25 mM Hepes and 2 mM EDTA) and stained for microglial markers.

Once single cell suspensions were obtained, the samples were blocked with anti-mouse CD16/CD32 (clone 93, 101321; BioLegend, 1:400 final concentration) for 10 min on ice and stained with anti-mouse CD11b-PECy7 (clone M1/70, 25-0112-82; eBioscience, 1:100 final concentration) and anti-mouse CD45-BV605 (clone 30-F11, 103139; BioLegend, 1:100 final concentration) for 30 min on ice. Staining antibodies were washed away with FACS buffer. Right before sorting, Draq7 (424001; BioLegend, 1:200 final concentration) was added to the cells to exclude dead cells. Sorting was performed on a BD Aria Fusion cell sorter equipped with four lasers: 405 nm violet (85 mW), 488 nm blue (50 mW), 561 nm yellow-green (50 mW), and 640 nm red (100 mW). Detection filters were 780/60 (LP735) for PE-CY7, 610/20(595LP) for BV605, and 780/60 (735LP) for DRAQ7. The cell suspension was sorted with an 85-$\mu$m nozzle at 45 PSI pressure. Cells were sequentially gated on FSC-A/FSC-H and SSC-A/SSC-H to select singlet events, followed by dead cell exclusion with DRAQ7. The final sort gate selected CD11b$^+$CD45$^{lo}$ events. Post-sort purity analysis included DRAQ5 (424101; BioLegend) for evaluating

---

dendritic process perimeter per Pu.1$^+$ nucleus. Only processes attached to nuclei were included in the dendritic process calculations. Statistics: two-way ANOVA (for genotype and age) and Tukey's *post hoc* test. Only select significant intergroup comparisons are shown. *$P < 0.05$; **$P < 0.01$; ***$P < 0.001$; ****$P < 0.0001$. Ctx cell number: genotype: $F_{(1,32)} = 4.892$, $P = 0.0342$; age: $F_{(1,32)} = 0.0001061$, $P = 0.9918$; interaction: $F_{(1,32)} = 0.61$, $P = 0.4405$. Ctx Iba1 area: genotype: $F_{(1,32)} = 8.461$, $P = 0.0065$; age: $F_{(1,32)} = 3.225$, $P = 0.0820$; interaction: $F_{(1,32)} = 2.182$, $P = 0.1494$. Ctx process area: genotype: $F_{(1,32)} = 9.908$, $P = 0.0035$; age: $F_{(1,32)} = 5.665$, $P = 0.0234$; interaction: $F_{(1,32)} = 3.165$, $P = 0.0847$. Ctx process perimeter: genotype: $F_{(1,32)} = 11.31$, $P = 0.0020$; age: $F_{(1,32)} = 2.868$, $P = 0.1000$; interaction: $F_{(1,32)} = 2.777$, $P = 0.1054$. Cbm cell number: genotype: $F_{(1,32)} = 4.982$, $P = 0.0327$; age: $F_{(1,32)} = 31.08$, $P < 0.0001$; interaction: $F_{(1,32)} = 1.119$, $P = 0.2817$. Cbm Iba1 area: genotype: $F_{(1,32)} = 2.602$, $P = 0.1165$; age: $F_{(1,32)} = 6.502$, $P = 0.0158$; interaction: $F_{(1,32)} = 0.6807$, $P = 0.4154$. Cbm process area: genotype: $F_{(1,32)} = 0.5592$, $P = 0.4601$; age: $F_{(1,32)} = 113$, $P < 0.0001$; interaction: $F_{(1,32)} = 2.322$, $P = 0.1374$. Cbm process perimeter: genotype: $F_{(1,32)} = 0.5271$, $P = 0.4731$; age: $F_{(1,32)} = 118.5$, $P < 0.001$; interaction: $F_{(1,32)} = 2.745$, $P = 0.1073$.

nucleated events; microglial purity was generally over 90% of nucleated events. The cells were maintained, chilled during the sort, and stored on ice. After sorting, the samples were spun down at 800*g* for 5 min to pellet the cells. The cell pellet was either processed right away for RNA isolation or frozen at −70°C.

For samples used for ChIP-seq, cells from three gender-matched animals were pooled together in one sample after staining and fixed in 1% paraformaldehyde for 10 min at room temperature, as described before (Gosselin et al, 2014). The cells were maintained in FACS buffer containing 1 mM sodium butyrate after fixation and after sorting. Cell pellets were frozen at −70°C until needed for ChIP-seq library preparation.

Peritoneal cells obtained from LPS-injected mice were not sorted. After washing with HBSS containing 25 mM Hepes, the cell pellets were processed for RNA isolation.

### RNA isolation and RNA-seq library preparation

The RNA (containing microRNAs) was isolated with the QIAGEN Universal AllPrep kit (80004; QIAGEN) according to the manufacturer's instruction. The RNA samples were treated on the column with DNAse I (79254; QIAGEN) to remove potential DNA contamination. The RNA concentration was quantified with the Quant-iT RiboGreen reagent (MP11490; Molecular Probes). The RNA quality was determined on a bioanalyzer with the RNA Pico 6000 kit (5067-1513; Agilent), according to the manufacturer's instructions.

Because of the low RNA content of microglia and our desire to use each animal as an individual sample, we used low RNA input protocols for RNA-seq library preparation. The cDNA for RNA-seq libraries was prepared using 10 ng of RNA using the SmartSeq v4 Ultra Low RNA Input Library preparation protocol (634889; Clontech) according to the manufacturer's instruction. The cDNA (250 ng starting material) was then tagged and fragmented with the NexteraXT DNA Library Preparation kit (FC-131-1024; Illumina) according to the manufacturer's instructions. Quality of amplified cDNA and fragmented libraries was assessed on a bioanalyzer with the High Sensitivity DNA chips (5067-4627; Agilent). Tagged libraries were pooled (8 nM total DNA) and sequenced on an Illumina HiSeq2500 sequencer, with a target depth of 20 million of 50 bp paired end reads. Quality control was performed using an Illumina's BaseSpace run summary tool.

Although microglial RNA samples for the aging study were obtained at different times, all libraries were prepared together to reduce batch effect in sequencing. It should be noted that the samples from 2-mo-old mice were used twice for sequencing—as a standalone set and as part of the aging study. The results from the two sequencing runs correlated very well with each other.

For LPS-injected mice, RNA from peritoneal cells (containing mostly peritoneal macrophages and B cells) was also subjected to sequencing, to confirm the peripheral effect of the LPS injections. The RNA-seq libraries were prepared with 8 ng RNA as the starting material for both peritoneal cells and matched microglia from the same animals to allow direct comparison of the cell types.

To validate the expression of selected genes by qRT-PCR, we sorted microglia from a separate cohort of mice and isolated RNA as described above. Fifty nanograms of RNA was reverse-transcribed to cDNA using the High-capacity cDNA Reverse Transcription kit (4368814; Thermo Fisher Scientific), which was then amplified with TaqMan Fast Advanced Master Mix (4444557; Thermo Fisher Scientific) on an Applied Biosystems QuantStudio 12K Flex thermocycler. The probes used were *Ccr1l1* (Mm00432606_s1), *H2-Aa* (Mm00439211_m1), *Klf2* (Mm00500486_g1), *S100a8* (Mm00496696_g1), *Egr1* (Mm00656724_m1), *Trem2* (Mm04209424_g1), and *Gapdh* (Mm99999915_g1). Gene expression within each sample was normalized to *Gapdh* and then to group average expression in WT mice. Samples are plotted individually and with their mean ± SEM.

### RNA-seq data analysis

The raw sequencing data were first analyzed using an in-house data-processing pipeline consisting of STAR (Dobin et al, 2013) and RSEM (Li & Dewey, 2014). Raw count files were aligned to mouse genome build mm10 using STAR and transcripts quantified with RSEM; in general, the unique mapping rate was >80%. The PCA was performed on the top 500 most variable genes after variance stabilizing transformation in DESeq2 (Love et al, 2014). Outliers were removed based on the first three principal components of the normalized and transformed count data. As illustrated in Fig 3B, three samples from LPS-injected mice (one WT, one Het, and one KO) shown in circles with a PC1 value around 25 clustered with saline-injected samples in squares, thus, were removed from subsequent analysis. The DEGs were calculated in DESeq2 by applying an additive model incorporating all the design factors (age, gender, and genotype) wherever applicable. Genes were considered significantly different between comparisons if they were expressed at greater than four transcripts per million mapped reads (TPMs) (average of all samples), they had at least 30% difference in fold change, and the false discovery rate (FDR) was less than 0.05. Analysis for each manipulation was carried out separately to avoid batch effects. Heatmap visualizations of the DEGs were made by using custom R scripts based on ComplexHeatmap (Gu et al, 2016). Overlap between gene sets was carried out with R VennDiagram package. Pathways enrichment analysis was performed with gene ontology as implemented in Illumina's BaseSpace Correlation Engine.

### ChIP-seq library preparation, sequencing, and data analysis

ChIP-seq libraries were prepared from recovered DNA by blunting, A-tailing, and adapter ligation using barcoded adapters (NextFlex; Bioo Scientific) as previously described (Heinz et al, 2010; Gosselin et al, 2014). Libraries were PCR-amplified for 12–15 cycles and size selected for fragments (225–400 bp) by gel extraction (10% Tris-boric acid-EDTA gels, EC62752BOX; Life Technologies). Libraries were single-end sequenced for 51 cycles, with an average sequence depth >20 million on an Illumina HiSeq 4000 or NextSeq 500 platform (Illumina) according to the manufacturer's instructions. The raw Rbp2 or H3K27Ac ChIP sequences were mapped to mm10 genome by Bowtie 2 (Langmead & Salzberg, 2013). Only uniquely mapped reads (>70%) were considered for downstream analyses. The peaks were called by HOMER (Heinz et al, 2010). The IDR (Li et al, 2011) was applied to extract confident peaks between replicates. Raw read numbers were quantified for peaks located 1,000 bp up- or down-stream of the transcription start site, and the peaks with less than four reads were removed. DESeq2 (Love

et al, 2014) was used to identify differential chromatin binding between *Cx3cr1*-KO and -WT mice. Binding/peaks were considered significantly different between comparisons if they had at least twofold difference in fold change and IDR less than 0.05. Two independent samples, each consisting of pooled microglia from three animals, were analyzed for each genotype.

## Analysis of microglial morphology

Young (2 mo) or aged (18–24 mo) $Cx3cr1^{+/+}$ or $Cx3cr1^{eGFP/eGFP}$ mice were euthanized by $CO_2$ asphyxiation, transcardially perfused with ice-cold PBS, and the brains were dissected out and cut into left and right hemisphere. One brain hemisphere was flash-frozen in liquid nitrogen, and the other was fixed in 10% neutral buffered formalin. After 48–72 h, formalin-fixed hemispheres were embedded in paraffin. Blocks were trimmed, and sections discarded until a level of 1.2 mm lateral of midline was reached. At that level, one section was collected for dual immunohistochemistry with Pu.1/Iba1 (Iba1: rabbit polyclonal; Wako, 0.25 μg/ml; Pu.1: rabbit monoclonal; Cell Signaling Technology, 5.96 μg/ml). Slides were scanned on a 3D-Histech Pannoramic-250 whole slide scanner and imported into Viosiopharm image analysis software. Tissue regions were manually annotated by a blinded analyst. A custom image analysis algorithm was designed to identify Pu.1-immunoreactive nuclei, their associated Iba1 immunoreactive microglial processes, and all other Iba1 immunoreactivity present in each tissue region. The cell soma were identified as Pu.1-positive when pixel intensity surpassed a minimum threshold. All contiguous Iba1 immunoreactivity within a 2-μm radius of a Pu.1-positive nucleus was defined as its microglial processes. The Iba1 immunoreactive area not directly attributed to a Pu.1-positive nucleus was defined as non-process Iba1. The computer-based algorithm quantified the areas, count, and perimeters of all identified cell features, for each brain region. Here, we report the number of Iba1$^+$/Pu.1$^+$ objects as the number of microglia per imaged area, the total Iba1-immunoreactive area per Pu.1$^+$ nucleus, the area of Iba1$^+$ processes attached to a Pu.1$^+$ nucleus as microglial process area, and the perimeter of Iba1$^+$ processed attached to a Pu.1$^+$ nucleus as the microglial process perimeter. In general, in young WT mice, a lower process area is associated with a more homeostatic phenotype, whereas an increased process area is associated with microglial activation.

The data were graphed by grouping animals by age and genotype; each point in the dot plots represents an individual animal. Statistical significance was determined by two-way ANOVA for the main effect of age or genotype (or their interaction), followed by pairwise comparisons between all groups with Tukey's *post hoc* test. Statistical significance on Tukey's test is indicated in the figures as *$P < 0.05$, **$P < 0.01$, ***$P < 0.001$, or ****$P < 0.0001$.

## Data availability

Raw sequencing files for the RNA-seq and ChIP-seq data sets are available at Gene Expression Omnibus (GSE131869 and GSE130960). Additional quality control plots and interactive data exploration tools built by QuickRNASeq pipeline (Zhao et al, 2016) are available for readers to generate boxplot, heatmap, and correlation plot on any genes of interest at https://baohongz.github.io/Cx3cr1-deficient-mice_microglia/. Please follow the user manual (http://baohongz.github.io/QuickRNASeq/guide.html) to take the full advantage of this shared resource including gene matrix tables in count, reads per kilobase of transcript per million mapped reads, and TPM measures and associated visualization widgets.

# Supplementary Information

## Author Contributions

S Gyoneva: conceptualization, data curation, formal analysis, visualization, methodology, and writing—original draft, review, and editing.
R Hosur: formal analysis, visualization, and writing—original draft.
D Gosselin: data curation, formal analysis, visualization, and writing—original draft.
B Zhang: software, formal analysis, visualization, and writing—review and editing.
Z Ouyang: formal analysis, visualization, and writing—original draft.
AC Cotleur: data curation, methodology, and writing—original draft.
M Peterson: data curation, formal analysis, visualization, and writing—original draft.
N Allaire: methodology.
R Challa: data curation, methodology, and writing—original draft.
P Cullen: data curation, methodology, and writing—original draft.
C Roberts: resources and supervision.
K Miao: data curation and writing—original draft.
TL Reynolds: formal analysis, supervision, visualization, and writing—original draft.
CK Glass: conceptualization, supervision, and writing—original draft.
L Burkly: conceptualization, supervision, and writing—original draft.
RM Ransohoff: conceptualization, supervision, and writing—original draft, review, and editing.

## Conflict of interest

Funding for this work was provided by Biogen as employee compensation (including stock options) or sponsored research agreement.

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
