## [Reviewer comments · Life Science Alliance]

Life Science Alliance

Cx3cr1-deficient microglia exhibit a premature aging transcriptome

Stefka Gyoneva, Raghavendra Hosur, David Gosselin, Baohong Zhang, Zhengyu Ouyang, Anne Coteleur, Michael Peterson, Norm Allaire, Ravi Challa, Patrick Cullen, Chris Roberts, Kelly Miao, Taylor Reynolds, Christopher Glass, Linda Burkly, and Richard Ransohoff

DOI: <https://doi.org/10.26508/lsa.201900453>

Corresponding author(s): Stefka Gyoneva, Biogen and Richard Ransohoff, Harvard Medical School

Review Timeline:

Submission Date:	2019-06-06
Editorial Decision:	2019-07-09
Revision Received:	2019-10-31
Editorial Decision:	2019-11-11
Revision Received:	2019-11-14
Accepted:	2019-11-17

Scientific Editor: Andrea Leibfried

Transaction Report:

July 9, 2019

Re: Life Science Alliance manuscript #LSA-2019-00453-T

Dr. Stefka Gyoneva
Biogen
Acute Neurology Research Unit
225 Binney St
Cambridge, MA 02142

Dear Dr. Gyoneva,

Thank you for submitting your manuscript entitled "Cx3cr1-deficient microglia exhibit a premature aging transcriptome" to Life Science Alliance. The manuscript was assessed by expert reviewers, whose comments are appended to this letter.

As you will see, both reviewers think that analyzing Cx3cr1-deficient cells is in principle providing a valuable resource since Cx3cr1 deficient mice are an important tool in the community. However, they also note, that claims pertaining specifically to microglia cannot be made and that there may be secondary effects due to the global KO. They further note some technical issues.

I have discussed your work further with an external advisor and concluded that we can offer to consider a revised version for publication here given the potential resource value. We would thus like to invite you to submit a revised version of your manuscript to us. Importantly, a revised version would have to avoid any over-interpretations regarding the usefulness of the model. The cutoff for the transcriptomics data analysis is not stringent enough and needs revisiting and candidate genes should get validated. Furthermore, histology would be essential to address reviewer #2's concerns.

Thank you for this interesting contribution to Life Science Alliance. We are looking forward to receiving your revised manuscript.

Sincerely,

B. MANUSCRIPT ORGANIZATION AND FORMATTING:

Reviewer #1 (Comments to the Authors (Required)):

Gyoneva et al. describe in their manuscript entitled "Cx3cr1-deficient microglia exhibit a premature aging transcriptome" a subset of immune genes that are dysregulated when the Cx3cr1 gene is knocked out in young mice and show that the expression of those genes is similar to wild-type aged mice, for which they used RNA-seq and histological techniques. The data remain descriptive, but are of interest, also since this gene is used very often for genetic modelling. These findings are indeed interesting but some clarifications on certain points are required, as well as improvements on some figures.

Major comments

1. Page 4, last paragraph: CX3CR1 is also expressed by CNS-associated macrophages (CAMs) and other myeloid cells as the authors also mention on page 16, paragraph 1. Isn't it then more correct to interpret the data by considering CNS-macrophages as a whole, rather than specifying microglia in particular, since the method is not able to distinguish between these cells?
2. Page 6, "In WT mice, the microglial expression profile obtained ...": Please provide direct experimental or analytical evidence for this statement.
3. Page 9, last sentence, Figure 5: In addition to unsupervised clustering: Generate aging signatures and CX3CR1 related signatures respectively and test them in the other dataset respectively.
4. Page 11, last paragraph: The authors use Iba1 for detecting microglia. However, it is known that Iba1 is not only specific to microglia. There is TMEM119, which is more specific to microglia than Iba1. Is there a reason of not choosing TMEM119?

Minor comments

1. General comment about figures: Using vector graphics in figures allows to zoom in to see details. Therefore, it is suggested that all figures (except images) are transformed into vector graphics. In particular, gene names displayed on the vertical axes of the heatmaps cannot be read.
2. Fig.1 legend: "Select genes" -> "Selected genes".
3. Fig. 1A: One dot in the Het group seems to be very different compared to the others of the same group (the one located at the lowest part of the figure). Is this real or due to experimental preparation? Please provide information on how pre-processing, outlier detection and quality control measure were performed during data processing.
4. Fig. 2A & B: Axis labels and marks have a bigger font size in fig.2A than in fig.2B. This should be made consistent.
5. Fig. 2C: The gene representations are too small and not readable.
6. Fig.3 legend: "select genes" -> "selected genes".
7. Fig. 3 A&B: It is difficult to see differences between KO and WT that are represented by the size of the markers. Using three different shapes for Het, KO and WT (for example circle, triangle and square), two different colors for MG/PC (red and blue, as it is already used) and two different fillings for LPS/Sal (filled shape vs empty shape with contour only), would make the figure easier to read. The size of the markers can then be adapted to decrease the overlap between markers by still keeping it in a reasonable visible size.
8. Sup. Fig. 2: Why is the color key used here different from those used in the other figures (for example sup. Fig. 1)? The color key goes from blue to red with a white intermediate instead of grey.

Reviewer #2 (Comments to the Authors (Required)):

Stefka Gyoneva et al

The authors performed RNA-Seq analysis on CD11b⁺/CD45^{low} cells collected from whole brain tissues from wild-type, Cx3cr1 heterozygous and Cx3cr1 knockout mice at 2 months, 1 year and 2 years of age. Independently of the genotype of the mouse, the authors report that the CD11b⁺/CD45^{low} cell population significantly decrease with aging (at 2 years ~1/3 of the population described at 2 months). They indicate that Cx3cr1 deficiency impacted on the transcriptome of these CD11b⁺/CD45^{low} cells. Indeed, using a cutoff of at least 30% increase in fold change for the analysis of their transcriptomic data, they list a number of differentially expressed genes that differ between the different mouse genotypes. Further, the authors suggest that the observed increase in transcripts for the majority of these differentially expressed genes is not due to transcriptional regulation, as they could not find association between RNA Polymerase II chromatin occupancies at their loci and expression of their messengers. The authors explored as well whether Cx3cr1 depletion can affect the transcriptional response of these CD11b⁺/CD45^{low} cells to a peripheral lipopolysaccharide challenge, but no significant effect as reported. Using unsupervised cluster analysis, they identified a pool of genes found to be expressed in CD11b⁺/CD45^{low} cells from both aged mice and Cx3cr1 deficiency mice even at 2 months of age. Based on this sole observation, the authors proposed that Cx3cr1 deletion in CD11b⁺/CD45^{low} cells promote an aged-like phenotype in those cells. Thereafter, the authors went on and investigated the impact of Cx3cr1 deficiency on IBA1- or PU1-expressing cells in different regions of the brain. In contrast to CD11b⁺/CD45^{low} cells, they report that the number IBA1-expressing cells was mostly not affected by aging, except for the cerebellum where their numbers were found to be increased. Finally, they report that in some brain region the morphology of these IBA-expressing cells was affected.

Overall, the collected data might be an interesting resource for the community. This is also a warning for the investigators using the Cx3cr1 mouse model. However, the interpretation of the data presented in this manuscript is problematic.

Major concerns.

The analysis of the genome wide transcriptomic data is done with an unusually low cutoff, i.e. 30% increase, which could significantly increase the number of false positive differentially expressed genes. There is an imperative need to validate those candidate genes by other methods.

The authors desire to draw conclusion on microglia, is not supported by their experimental data. They make the assumption that independently of age, genotype of the mice, microglia can be collected from whole brain tissue based on CD11b and CD45^{low} expression. This is not taking into account the reports that CD45 expression can be regulated in microglia. Their own data illustrate the caveat of their investigation, whereas CD11b⁺/CD45^{low} cells decrease with aging, IBA1⁺ or PU1⁺ cells did not. Of note, both approaches were used by the authors to investigate microglia. These data suggest that they are looking at a subpopulation of microglia but conclusions are made on microglia globally.

The authors proposed that the difference in transcriptome between the different Cx3cr1 genotypes might not be due to transcriptional regulation per se. This statement should be supported by experimental data.

The authors propose that Cx3cr1 deficiency promote an accelerated aging of microglia. This should be supported by experimental data, and not only rely on the presented gene cluster analysis.

Finally, as the authors mentioned the use of germline and globally deficient Cx3cr1 mice is indeed a

concern when microglia are to be investigated, are non-microglia related effects are expected.

Responses to reviewers' comments

Gyoneva et al., *Cx3cr1-deficient microglia exhibit a premature aging transcriptome*

Dear Editor and Reviewers,

During the course of manuscript preparation, there was a change in analysisist for the RNA-seq data, with R.H. and C.R. moving on to new positions and B.Z. coming on board. In response to the reviewers' comments, B.Z. updated the RNA-seq analyses to be consistent with current standards in the field, which included updating how the data were normalized and visualized. Specifically,

- Previous PCA analysis was performed on normalized counts that biases the analysis to genes with large variance and highest counts because they show the largest absolute differences between samples, which in turn subdues less dominant principle components. The current analysis use variance-stabilizing transformation implemented in DESeq2 that resembles normal distribution better and is less sensitive to high count outliers than both untransformed normalized counts and plain log transformation so that the 2nd or 3rd principle components representing genotype and gender differences are showing up.
- The current heatmaps use Z-scores instead of log2 of gene expression measurements since Z-scores are centered and normalized, so the reader can interpret a color as x standard deviations from the mean and have an intuitive idea of the relative variation of gene expression across samples.

The Materials and Methods section has been updated to reflect the new methods used. The new analysis did not change the main conclusions of the manuscript about the modulating of aging-related pathways by *Cx3cr1* deletion. Instead, the new analysis revealed a stronger gender effect than previously appreciated, which we have now incorporated into the manuscript. Finally, the updated version of the manuscript could be published with a portal that contains all scripts used to generate the analyses and figures. For his work, B.Z. is now added as a co-author.

Responses to the reviewers' specific comments are below.

Reviewer #1 (Comments to the Authors (Required)):

Gyoneva et al. describe in their manuscript entitled "Cx3cr1-deficient microglia exhibit a premature aging transcriptome" a subset of immune genes that are dysregulated when the Cx3cr1 gene is knocked out in young mice and show that the expression of those genes is similar to wild-type aged mice, for which they used RNA-seq and histological techniques. The data remain descriptive, but are of interest, also since this gene is used very often for genetic modelling. These findings are indeed interesting but some clarifications on certain points are required, as well as improvements on some figures.

Major comments

1. Page 4, last paragraph: *CX3CR1* is also expressed by CNS-associated macrophages (CAMs) and other myeloid cells as the authors also mention on page 16, paragraph 1. Isn't it then more correct to interpret the data by considering CNS-macrophages as a whole, rather than specifying microglia in particular, since the method is not able to distinguish between these cells?
 - a. **Reply:** CNS-associated macrophages (CAMs) indeed express *CX3CR1*. For our transcriptomic analysis, we isolated $CD11b^+CD45^{lo}$ cells. Because CAMs are $CD45^{hi}$, they are not included in the transcriptomic analysis.
With this said, we cannot rule out that the transcriptional changes we identify in $CD11b^+CD45^{lo}$ cells may be a consequence of *Cx3cr1* loss in another cell type. We acknowledge this possibility on page 16.
 - b. **Responsive changes to the text (p.17):** "Although cellular expression of *Cx3cr1* is highest for microglia, there is also robust *Cx3cr1* expression by varied subsets of peripheral immune cells and other CNS-associated macrophages (Jung *et al.*, 2000). ... In light of these considerations, the microglial transcriptomic phenotypes observed in *Cx3cr1*-deficient mice could be governed in part by loss of peripheral *Cx3cr1* or loss of *Cx3cr1* in the CNS-associated macrophages."
2. Page 6, "In WT mice, the microglial expression profile obtained ...": Please provide direct experimental or analytical evidence for this statement.
 - a. **Reply:** We compiled a spreadsheet with the top 50 highest expressed genes identified in this study, as well as the top 50 reported by Zhang *et al.* and Hickmann *et al.* (microglial sensome enriched to brain), and the microglia signature reported by Butovsky *et al.* These lists are now available as Supplementary Table 1. The remaining supplementary tables have been renumbered.
 - b. **Responsive changes to the text (p. 6):** "In WT mice, the microglial expression profile obtained here is consistent with published microglial transcriptomes (Supplementary Table 1) (Hickman *et al.*, 2013; Butovsky *et al.*, 2014; Zhang *et al.*, 2014)."
3. Page 9, last sentence, Figure 5: In addition to unsupervised clustering: Generate aging signatures and *CX3CR1* related signatures respectively and test them in the other dataset respectively.
 - a. **Reply:** As the reviewer suggested, we performed an additional unsupervised clustering analysis, using top 200 DEGs between WT 2 yr vs 2 mos microglia. The samples clearly cluster by age. First, genes in the pathways regulated by *Cx3cr1* loss, namely MHCII genes, chemokines and chemokine receptors, inflammatory genes in general, are also regulated by age. Second, although the clustering by genotype is not as obvious within this aging transcriptome, there appears to be a grouping of WT and Het samples in 2 mos mice, which is lost on the aged mice. We have included these new data as Supplementary Figure 4.
 - b. **Responsive changes to the text (p. 11, Results):** We also performed unsupervised cluster analysis using the top 200 DEGs between WT 2 yr vs 2 mos

microglia rather than the genotype-regulated DEGs (Supplementary Figure 4). Using this gene list, the samples clearly cluster by age. Genes in the pathways regulated by *Cx3cr1* loss, namely MHCII genes, chemokines and chemokine receptors, inflammatory genes in general, are also regulated by age, as has been shown before (Grabert *et al.*, 2016). Moreover, although the clustering by genotype is not as obvious within this aging transcriptome, there appears to be a grouping of WT and Het samples in 2 mos mice, which is lost on the aged mice. These results support our finding that *Cx3cr1* loss and aging modulate the expression of similar genes and pathways.

4. *Page 11, last paragraph: The authors use Iba1 for detecting microglia. However, it is known that Iba1 is not only specific to microglia. There is TMEM119, which is more specific to microglia than Iba1. Is there a reason of not choosing TMEM119?*
 - a. **Reply:** Iba1 is the gold standard marker for microglia being both sensitive and specific for identifying microglia in the brain parenchyma. TMEM119, recently described, does not stain all parenchymal microglia (Fig 1-2 *Neuron*. 2018 Jun 27; 98(6): 1170–1183.e8.). In the brain parenchyma, the only non-microglial Iba1-positive cells would be infiltrating monocytes, which would be a very rare occurrence in the healthy brain and would have negligible/no contribution to our morphology readouts. To avoid misrepresentation in the description of the morphology results, we replaced some references to “microglia” with “Iba1 positive cells.”
 - b. **Responsive changes to the text** (p. 19, Discussion): “We extended our characterization of the effects of *Cx3cr1* deletion on microglia by analyzing the morphology of Iba1-immunoreactive cells (Figure 7). Microglia in the cortex, striatum and hippocampus of young mice showed reduced ramification assessed by reduced Iba1 process area and perimeter in *Cx3cr1*-KO mice compared to -WT mice (Figure 7A, C; Supplementary Figure 5).”

Minor comments

1. *General comment about figures: Using vector graphics in figures allows to zoom in to see details. Therefore, it is suggested that all figures (except images) are transformed into vector graphics. In particular, gene names displayed on the vertical axes of the heatmaps cannot be read.*
 - a. **Reply and responsive changes to the text:** We apologize that the quality of the figures in the original submission was suboptimal, making it difficult to the reviewers to read them. We regenerated all RNA-seq-based figures using vector graphics and uploaded high-resolution images for review. In addition, we have created a portal where the RNA-seq data can be queried and visualized by the reviewers and future readers. A link to the portal is included in the Data Availability section (p.27).
2. *Fig.1 legend: "Select genes" -> "Selected genes".*
 - a. **Reply and responsive changes to the text:** The text has been corrected.

3. *Fig. 1A: One dot in the Het group seems to be very different compared to the others of the same group (the one located at the lowest part of the figure). Is this real or due to experimental preparation? Please provide information on how pre-processing, outlier detection and quality control measure were performed during data processing.*
 - a. **Reply and responsive changes to the text:** Outliers were not removed during the RNAseq processing pipeline, unless a biological reason was identified (e.g., LPS-treated animals did not show signs of inflammation). The QC data for all samples is available on the RNAseq portal mentioned above for the reviewers and readers to peruse.
4. *Fig. 2A & B: Axis labels and marks have a bigger font size in fig.2A than in fig.2B. This should be made consistent.*
 - a. **Reply and responsive changes to the text:** We thank the reviewer for noticing this. The fonts were adjusted to make them consistent.
5. *Fig. 2C: The gene representations are too small and not readable.*
 - a. **Reply and responsive changes to the text:** We re-generated the figure. The new version should allow zooming to read the gene names.
6. *Fig.3 legend: "select genes" -> "selected genes".*
 - a. **Reply and responsive changes to the text:** The text has been corrected.
7. *Fig. 3 A&B: It is difficult to see differences between KO and WT that are represented by the size of the markers. Using three different shapes for Het, KO and WT (for example circle, triangle and square), two different colors for MG/PC (red and blue, as it is already used) and two different fillings for LPS/Sal (filled shape vs empty shape with contour only), would make the figure easier to read. The size of the markers can then be adapted to decrease the overlap between markers by still keeping it in a reasonable visible size.*
 - a. **Reply and responsive changes to the text:** Based on the reviewer's suggestion, we re-evaluated our use of symbols throughout the manuscript. For figure 3B, we decided to use shape for treatment, color for genotype and open or filled symbol for gender. In addition, we tried to keep these designations as consistent as possible for other figures.
8. *Sup. Fig. 2: Why is the color key used here different from those used in the other figures (for example sup. Fig. 1)? The color key goes from blue to red with a white intermediate instead of grey.*
 - a. **Reply and responsive changes to the text:** The color key was different in this figure because it used a different software package/user to generate the heatmap. All heatmaps were regenerated for this resubmission to ensure consistent formatting.

Reviewer #2 (Comments to the Authors (Required)):

Stefka Gyoneva et al

The authors performed RNA-Seq analysis on CD11b⁺/CD45^{low} cells collected from whole brain tissues from wild-type, Cx3cr1 heterozygous and Cx3cr1 knockout mice at 2 months, 1 year and 2 years of age. Independently of the genotype of the mouse, the authors report that the CD11b⁺/CD45^{low} cell population significantly decrease with aging (at 2 years ~1/3 of the population described at 2 months). They indicate that Cx3cr1 deficiency impacted on the transcriptome of these CD11b⁺/CD45^{low} cells. Indeed, using a cutoff of at least 30% increase in fold change for the analysis of their transcriptomic data, they list a number of differentially expressed genes that differ between the different mouse genotypes. Further, the authors suggest that the observed increase in transcripts for the majority of these differentially expressed genes is not due to transcriptional regulation, as they could not find association between RNA Polymerase II chromatin occupancies at their loci and expression of their messengers. The authors explored as well whether Cx3cr1 depletion can affect the transcriptional response of these CD11b⁺/CD45^{low} cells to a peripheral lipopolysaccharide challenge, but no significant effect as reported. Using unsupervised cluster analysis, they identified a pool of genes found to be expressed in CD11b⁺/CD45^{low} cells from both aged mice and Cx3cr1 deficiency mice even at 2 months of age. Based on this sole observation, the authors proposed that Cx3cr1 deletion in CD11b⁺/CD45^{low} cells promote an aged-like phenotype in those cells. Thereafter, the authors went on an investigated the impact of Cx3cr1 deficiency on IBA1- or PU1-expressing cells in different regions of the brain. In contrast to CD11b⁺/CD45^{low} cells, they report that the number IBA1-expressing cells was mostly not affected by aging, expect for the cerebellum where their numbers were found to be increased. Finally, they report that in some brain region the morphology of these IBA-expressing cells was affected.

Overall, the collected data might be an interesting resource for the community. This is also a warning for the investigators using the Cx3cr1 mouse model. However, the interpretation of the data presented in this manuscript is problematic.

Major concerns.

- 1. The analysis of the genome wide transcriptomic data is done with an unusually low cutoff, i.e. 30% increase, which could significantly increase the number of false positive differentially expressed genes. There is an imperative need to validate those candidate genes by other methods. Histology?*
 - Reply:** Although RNAseq analysis has significantly advanced over the years to yield reliable data, we validated selected genes by qPCR, which is more sensitive than histology. Using a separate cohort of WT, Het and KO mice, we sorted microglia as described in the Materials and Methods section, isolated RNA, and performed targeted qRT-PCR. The probe information is listed in the Materials and Methods section. The validation data is included as Supplementary Figure 1 (and the rest of the figures have been re-numbered). In general, we confirmed the effect of Cx3cr1 genotype on the expression of key genes identified by qPCR.

b. **Responsive changes to the text:**

- i. Added to p. 7 (Results): Although fewer genes had higher expression in *Cx3cr1*-deficient microglia, some of these included genes with pleiotropic functions such as *Tnf*, the transcription factors *Egr1* and *Klf2*, and the chemokine-like receptor *Ccr1/1*. We confirmed the altered expression of selected genes by *Cx3cr1* genotype by qRT-PCR using microglia sorted from a separate cohort of mice (Supplementary Figure 1A), but not others (*Egr1*).
- ii. Added to p. 25 (Materials and Methods): To validate the expression of selected genes by qRT-PCR, we sorted microglia from a separate cohort of mice and isolated RNA as described above. Fifty nanograms of RNA was reverse-transcribed to cDNA using the High-Capacity cDNA Reverse Transcription kit (ThermoFisher 4368814), which was then amplified with TaqMan™ Fast Advanced Master Mix (ThermoFisher 4444557) on an Applied Biosystems QuantStudio™ 12K Flex thermocycler. The probes used were: *Ccr1/1* (Mm00432606 s1), *H2-Aa* (Mm00439211 m1), *Klf2* (Mm00500486 g1), *S100a8* (Mm00496696 g1), *Egr1* (Mm00656724 m1), *Trem2* (Mm04209424 g1), and *Gapdh* (Mm99999915 g1). Gene expression within each sample was normalized to *Gapdh* and then to group average expression in WT mice. Samples are plotted individually and with their mean +/- SEM.

2. *The authors desire to draw conclusion on microglia, is not supported by their experimental data. They make the assumption that independently of age, genotype of the mice, microglia can be collected from whole brain tissue based on CD11b and CD45^{low} expression. This is not taking into account the reports that CD45 expression can be regulated in microglia. Their own data illustrate the caveat of their investigation, whereas CD11b⁺/CD45^{low} cells decrease with aging, IBA1⁺ or PU1⁺ cells did not. Of note, both approaches were used by the authors to investigate microglia. These data suggest that they are looking at a subpopulation of microglia but conclusions are made on microglia globally.*

- a. **Reply:** The reviewer raises a very good point with which we agree. Indeed, it is possible that the transcriptomes we identified are specific to the microglia recovered by the cell sorting procedure and may not be reflective of all microglia in the brain. We have now acknowledged this technical concern in the Discussion. At the same time, the agreement between RNAseq and histology, namely that *Cx3cr1*-deficient microglia may have a reduced expression of inflammatory components of the microglial transcriptome, suggest that at least under some conditions, the transcriptomes of sorted cells may be generalizable to the overall microglial population.
- b. **Responsive changes to the text** (p.17): Moreover, we cannot rule out the possibility that loss of *Cx3cr1* may affect our ability to recover microglia from all conditions with equal efficiency. As such, the transcriptomes we report here may not be generalizable to the whole microglial population *in situ* in the brain. Yet,

both histology (Figure 7) and RNA-seq analysis (Figure 1) suggest that microglia from *Cx3cr1*-deficient animals show a reduced inflammatory response, supporting the validity of our observations.

3. *The authors proposed that the difference in transcriptome between the different *Cx3cr1* genotypes might not be due to transcriptional regulation per se. This statement should be supported by experimental data.*
 - a. **Reply:** Our ChIP-Seq data, especially with Rbp2, support this observation because Rbp2 is expected to bind only at chromatin regions undergoing active transcription. This has been clarified in the text. If the reviewer is suggesting that post-transcriptional mechanisms of gene regulation should be explored, we note that such studies lie outside the scope of the present report.
 - b. **Responsive changes to the text** (p.8): We performed chromatin immunoprecipitation for RNA polymerase II (Rbp2) followed by sequencing (ChIP-Seq) to determine whether changes in transcriptional activity indicated by Rbp2 binding explain the differential gene expression between *Cx3cr1*-WT and -KO microglia.
4. *The authors propose that *Cx3cr1* deficiency promote an accelerated aging of microglia. This should be supported by experimental data, and not only rely on the presented gene cluster analysis.*
 - a. **Reply:** The study suggested by the reviewer is an important next step to our research and of high interest to us, but it is beyond the scope of the current manuscript.
5. *Finally, as the authors mentioned the use of germline and globally deficient *Cx3cr1* mice is indeed a concern when microglia are to be investigated, are non-microglia related effects are expected.*
 - a. **Reply and responsive changes to the text:** It is plausible that peripheral *Cx3cr1*-positive cells may show altered transcriptomes in response to *Cx3cr1* deletion. In the Discussion, as the reviewer noted, we acknowledge that changes to peripheral cells may affect the microglial transcriptome. Although we do not expand on it in the current manuscript, we observed that germline loss of *Cx3cr1* did not affect the transcriptomes of peritoneal cells, which are *Cx3cr1*-negative.

November 11, 2019

RE: Life Science Alliance Manuscript #LSA-2019-00453-TR

Dr. Stefka Gyoneva
Biogen
Acute Neurology Research Unit
225 Binney St
Cambridge, MA 02142

Dear Dr. Gyoneva,

Thank you for submitting your revised manuscript entitled "Cx3cr1-deficient microglia exhibit a premature aging transcriptome". As you will see, reviewer #2 re-assessed your work and appreciates the introduced changes, and we would thus be happy to publish your paper in Life Science Alliance pending final revisions necessary to meet our formatting guidelines:

- One of the previous co-authors got removed from the submission; please either fix or have all authors, including the author that got removed, send us a message confirming that this author change is fine
- All corresponding authors should please link their profiles in our submission system to their ORCID IDs; you should have received an email with instructions on how to do so
- You mention Fig. 1D in the manuscript text, but there is no panel D in Fig. 1
- Please list 10 authors et al in your reference list
- I think the scale bars currently used in the tissue sections may confuse some readers, please use a solid line instead. It is OK to display the scale bar only on one image / panel since the scale remains the same

A. FINAL FILES:

-- High-resolution figure, supplementary figure and video files uploaded as individual files: See our

detailed guidelines for preparing your production-ready images, <http://www.life-science-alliance.org/authors>

B. MANUSCRIPT ORGANIZATION AND FORMATTING:

Sincerely,

Andrea Leibfried, PhD
Executive Editor
Life Science Alliance
Meyerhofstr. 1
69117 Heidelberg, Germany
t +49 6221 8891 502

e.a.leibfried@life-science-alliance.org
www.life-science-alliance.org

Reviewer #2 (Comments to the Authors (Required)):

The revised version of the manuscript addressed the reviewer's comment in a satisfactory manner.
The data presented are of importance for the field.

November 17, 2019

RE: Life Science Alliance Manuscript #LSA-2019-00453-TRR

Dr. Stefka Gyoneva
Biogen
Acute Neurology Research Unit
225 Binney St
Cambridge, MA 02142

Dear Dr. Gyoneva,

Thank you for submitting your Research Article entitled "Cx3cr1-deficient microglia exhibit a premature aging transcriptome". It is a pleasure to let you know that your manuscript is now accepted for publication in Life Science Alliance. Congratulations on this interesting work.

DISTRIBUTION OF MATERIALS:

Again, congratulations on a very nice paper. I hope you found the review process to be constructive and are pleased with how the manuscript was handled editorially. We look forward to future exciting submissions from your lab.

Sincerely,
